METHODS AND RESOURCES

# MycoRed: Betalain pigments enable in vivo real-time visualisation of arbuscular mycorrhizal colonisation

**Alfonso Timoneda**[1☯], **Temur Yunusov**[2☯], **Clement Quan**[2], **Aleksandr Gavrin**[2], **Samuel F. Brockington**[1☯]*, **Sebastian Schornack**[2☯]*

**1** Department of Plant Sciences, University of Cambridge, Cambridge, United Kingdom, **2** Sainsbury Laboratory, University of Cambridge, Cambridge, United Kingdom

☯ These authors contributed equally to this work.
* sb771@cam.ac.uk (SFB); sebastian.schornack@slcu.cam.ac.uk (SS)

**Data Availability Statement:** All relevant data are within the paper and its Supporting Information files.

## Abstract

Arbuscular mycorrhiza (AM) are mutualistic interactions formed between soil fungi and plant roots. AM symbiosis is a fundamental and widespread trait in plants with the potential to sustainably enhance future crop yields. However, improving AM fungal association in crop species requires a fundamental understanding of host colonisation dynamics across varying agronomic and ecological contexts. To this end, we demonstrate the use of betalain pigments as in vivo visual markers for the occurrence and distribution of AM fungal colonisation by *Rhizophagus irregularis* in *Medicago truncatula* and *Nicotiana benthamiana* roots. Using established and novel AM-responsive promoters, we assembled multigene reporter constructs that enable the AM-controlled expression of the core betalain synthesis genes. We show that betalain colouration is specifically induced in root tissues and cells where fungal colonisation has occurred. In a rhizotron setup, we also demonstrate that betalain staining allows for the noninvasive tracing of fungal colonisation along the root system over time. We present MycoRed, a useful innovative method that will expand and complement currently used fungal visualisation techniques to advance knowledge in the field of AM symbiosis.

## Introduction

Arbuscular mycorrhiza (AM) fungi of the subphylum Glomeromycotina are soil fungi that engage in symbiosis with land plants [1]. Symbiotic associations with AM fungi date back to over 400 million years ago and can be formed by 70% to 72% of extant land plant species [2–4]. AM fungi are obligate biotrophs that receive all their carbon intake from the plant, which is estimated at up to 20% of the plant's photosynthate [5]. In exchange, the fungus assists the plant with the acquisition of mineral nutrients, mainly phosphorus, whose availability in soils is often a limiting factor for plant growth [6]. Phosphorus contribution through the mycorrhizal pathway can be very high, and, in some instances, can account for the entire phosphorus consumption of a plant [7]. During AM symbiosis, fungal hyphae form dichotomously branched structures, named arbuscules, within root cortex cells. Hyphal extension and

**Funding:** This work was supported by an iCASE BBSRC-DTP (RG88096) sponsored by Coca-Cola to S.F.B & A.T., Gatsby Charitable Foundation (GAT3395/GLD) and Royal Society (UF110073, UF160413) to S.S., and a Natural Environmental Research Council Independent Research Fellowship (NE/K009303) to S.F.B. The funders had no role in study design, data collection and analysis, decision to publish, or preparation of the manuscript.

**Competing interests:** The authors have declared that no competing interests exist.

**Abbreviations:** AM, arbuscular mycorrhiza; BCP, blue copper protein; BIA, benzylisoquinoline alkaloid; cDOPA5GT, *cyclo*-DOPA 5-*O*-glucosyltransferase; *cyclo*-DOPA, *cyclo*-dihydroxyphenylalanine; *dmi3, Doesn't Make Infections 3*; DODA, ʟ-DOPA-4,5-dioxygenase; dpi, days postinoculation; ʟ-DOPA, 3,4-dihydroxy-ʟ-phenylalanine; PHT1, phosphate transporter 1; QTL, quantitative trait locus; *RiBTub, R. irregularis* β-tubulin; RT-PCR, reverse transcription PCR; WGA, wheat germ agglutinin; WT, wild-type; X-gluc, 5-bromo-4-chloro-3-indoxyl-β-D-glucuronid acid.

arbuscule formation are accompanied by the de novo extension of a specialised plant cell membrane that separates the fungal hyphae from the plant cytoplasm [8]. In order to accommodate arbuscule formation, plant cells undergo a series of changes in gene expression to aid the establishment of symbiosis [9]. Examples of such AM-induced genes include *MtPT4* and *MtBCP1* in the model legume *Medicago truncatula*. *MtPT4* encodes a phosphate transporter, belonging to the phosphate transporter 1 (PHT1) subfamily, which is exclusively expressed in arbuscule-containing cells [10]. *Mt*PT4 localises to the plant cell membrane surrounding arbuscules and participates in the acquisition of phosphate released by the fungus during symbiosis. *MtBCP1* encodes a member of the blue copper protein (BCP) family and is also specifically expressed in regions of the root hosting arbuscule development during AM symbiosis [11]. *MtBCP1* expression is strongest in arbuscule-containing cells but can additionally be observed in adjacent cortical cells [11].

The study of AM symbiotic processes involves the detection, visualisation, and quantification of fungal colonisation. Current techniques rely on the specific staining of fungal cell walls through fast, simple, and cost-effective procedures. Commonly employed methods include the use of trypan blue [12], cotton blue and Sudan IV [13], acid fuchsin [14], ink–vinegar [15], or fluorescein-labelled wheat germ agglutinin (WGA) [16]. All of these methods are destructive, requiring the excision and chemical treatment of roots that are typically visualised with light or fluorescent microscopy. On the other hand, nondestructive methods for detection and visualisation of AM symbiosis offer a number of significant research opportunities but often rely on specialised equipment or are only available for certain species. For example, the foliar accumulation of blumenol-derived metabolites can function as a quantitative proxy for AM colonisation in a number of crop and model plants, with potential applications in field-based quantitative trait locus (QTL) mapping of AM fungi-related genes [17]. However, blumenol accumulation is not visible, and its detection requires specialised extraction and quantification steps. Furthermore, some cereal crops and species of the Liliaceae and Fabaceae naturally produce apocarotenoid yellow pigments in roots upon mycorrhizal colonisation [18,19], which has been useful, for example, for the identification of maize mutants affected in symbiotic interaction [20]. To date, however, the use of natural pigments as visual markers of AM symbiosis is limited to select species within these families and has not been implemented in other crops and model plants.

Betalains are naturally occurring tyrosine-derived water-soluble pigments, comprising yellow to orange betaxanthins and red to purple betacyanins. Betalains have a number of bioindustrial applications, as natural food colourants and antioxidants [21], and as biosensors [22–26]. Betalains were first discovered in plants, where they are unique to the flowering plant order Caryophyllales [27], but have also been reported in the proteobacterium *Gluconacetobacter diazotrophicus* [28] and the fungi *Amanita* [29] and *Hygrocybe* [30]. The betalain biosynthesis pathway in Caryophyllales consists only of 3 main enzymatic steps with additional glycosylation and spontaneous chemical reactions (Fig 1). Initially, tyrosine undergoes an hydroxylation reaction to 3,4-dihydroxy-ʟ-phenylalanine (ʟ-DOPA) catalysed by members of the CYP76AD family of cytochrome P450 enzymes (CYP76AD1/5/6 in *Beta vulgaris*) [31,32]. ʟ-DOPA is then cleaved by the action of the ʟ-DOPA-4,5-dioxygenase (DODA) enzyme to form a 4,5-secodopa intermediate that spontaenously cyclises to betalamic acid [33]. Betalamic acid is the central betalain chromophore and can spontaneously condense with amino groups to produce yellow betaxanthins [34]. Alternatively, betalamic acid can spontaneously condense with *cyclo*-dihydroxyphenylalanine (*cyclo*-DOPA) to produce the red betacyanin—betanidin. *Cyclo*-DOPA is obtained via oxidation of ʟ-DOPA, catalysed by CYP76AD1 [35]. Glycosylation of betacyanins is common and can occur at either the betanidin [36] or *cyclo*-DOPA

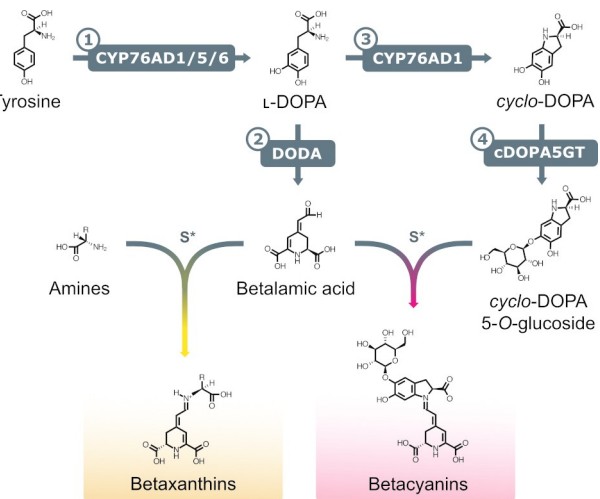

**Fig 1. The betalain biosynthetic pathway.** A simplified schematic representation of the main enzymatic and spontaneous reactions leading to the formation of red/purple betacyanins and yellow betaxanthins. Enzymatic steps: (1) tyrosine hydroxylation to ʟ-DOPA catalysed by CYP76AD cytochrome P450 enzymes; (2) cleavage of ʟ-DOPA to form betalamic acid by DODA; (3) ʟ-DOPA oxidation to *cyclo*-DOPA by CYP76AD1; and (4) *cyclo*-DOPA glycosylation to cyclo-DOPA 5-*O*-glucoside by the enzyme cDOPA5GT. S*, spontaneous reaction. Betacyanins are represented as a molecule of betanin. cDOPA5GT, *cyclo*-DOPA 5-*O*-glucosyltransferase; *cyclo*-DOPA, *cyclo*-dihydroxyphenylalanine; DODA, ʟ-DOPA-4,5-dioxygenase; ʟ-DOPA, 3,4-dihydroxy-ʟ-phenylalanine.

stages [37], with the latter catalysed by the enzyme *cyclo*-DOPA 5-*O*-glucosyltransferase (cDO-PA5GT). Betanin is the glycosylated form of betanidin.

Elucidation of these core enzymatic steps has enabled the engineering of the betalain biosynthetic pathway in a wide range of heterologous hosts, including microbes such as *Saccharomyces cerevisiae* [38]; plant model organisms like *Arabidopsis thaliana*, *Nicotiana tabacum*, and *Petunia hybrida* [31,39,40]; and a diversity of crops such as *Oryza sativa* (rice), *Solanum lycopersicum* (tomato), *Solanum tuberosum* (potato), and *Solanum melongena* (aubergine) [40,41]. Betalains have been used as biosensors in a number of heterologous contexts to report increased production of metabolites, such as tyrosine, dopamine, and ʟ-DOPA in *Escherichia coli* and *Nicotiana benthamiana* [23,24,26], to measure metabolic flux between competing pathways in the synthesis of benzylisoquinoline alkaloids (BIAs) in *S. cerevisiae* [22] and for the detection of copper by heavy metal–resistant bacteria in bioremediation processes [25]. In plants, specific promoters have been successfully used to target betalain production in specific tissues such as fruits and seed endosperm [40–42]. Expressing the betalain biosynthetic genes under the control of the *AtYUC4* promoter in *A. thaliana* resulted in pigment production in tissues likely to present auxin biosynthesis activity [42]. Similarly, the use of the DR5 synthetic auxin-responsive promoter in *O. sativa* calli allowed for easier selection of transformed calli in in vitro transformation protocols [42]. The use of betalains as in vivo reporters offer a number of advantages: (1) the relative simplicity of betalain biosynthesis; (2) the potential for heterologous betalain expression in phylogenetically diverse hosts; and (3) the ease of betalain visualisation and quantification.

Here, we present MycoRed, a betalain-based in vivo and noninvasive reporter system for the occurrence and progression of AM symbiosis in roots of the two model species *M. truncatula* and *N. benthamiana*. We leveraged known AM-responsive genes from *M. truncatula* to identify orthologous *N. benthamiana* promoters that are similarly responsive to AM fungal colonisation. Heterologous expression of betalain biosynthesis genes specifically driven by AM-responsive promoters effectively tracked AM colonisation dynamics in both species.

Collectively, our work demonstrates the efficacy of betalain pigments as reliable in vivo visual markers for the previously inaccessible dynamic tracing of AM symbiosis within root systems, thereby providing a valuable resource for the plant–microbe research community.

## Results

### Betalains can be used to visualise AM fungus colonisation in living *Medicago truncatula* roots

To establish a reporter system that allows for the noninvasive visualisation of AM fungal colonisation in roots, we explored the use of betalain biosynthesis genes under the control of AM symbiosis–specific plant promoters. In *M. truncatula*, *PT4* and *BCP1* are specifically expressed during root colonisation by AM fungi and are often used as markers to quantify the extent of AM symbiosis [10,11]. We generated T-DNA constructs harbouring CYP76AD1, the enzyme catalysing steps 1 and 3 of the betalain biosynthetic pathway (Fig 1), driven by either *MtPT4* or *MtBCP1* symbiotic-specific promoters.The T-DNA also carried genes encoding the two additional enzymes needed to produce betalains: *DODA* and *cDOPA5GT* (steps 2 and Fig 1) under *35S* and *Ubi10* constitutive promoters, respectively. We decided to use different promoters in order to avoid transcriptional gene silencing induced by sequence repeat. We refer to these multigene vectors as *MtPT4*-p1 and *MtBCP1*-p1 (Fig 2A). Next, we generated *M. truncatula* composite plants expressing these constructs in hairy roots. Resultant *MtPT4*-p1 and *MtBCP1*-p1 transgenic roots did not display visible pigments prior to AM inoculation. After 4 weeks of colonisation with the AM fungus *Rhizophagus irregularis*, AM symbiosis marker genes were transcriptionally induced, and the roots produced visibly red-coloured root (Figs 2B–2G and S1). We did not observe any pigment production or AM marker gene induction in mock conditions for *MtPT4*-p1 or *MtBCP1*-p1 composite roots. Dissection and microscopy of pigmented and nonpigmented root fragments revealed that betacyanin production consistently colocalised or was adjacent to arbuscule-containing cells both in *MtPT4*-p1 and *MtBCP1*-p1 roots (Fig 3). Pigment presence often extended beyond arbuscule-containing cells in the inner cortex and was also observed in the endodermis, the pericycle, the stele, and, in some cases, adjacent cortical cells, which represent root tissue layers that did not show any intracellular fungal structures (Fig 3). To confirm that the employed promoter fragments were indeed associated with intracellular fungal structures, we generated *MtPT4* and *MtBCP1* GUS fusions and expressed them in roots of *M. truncatula* composite plants (S2 Fig). Staining pattern after fungal colonisation showed that prominent GUS substrate accumulation was limited to inner cortical cells colonised by the fungus, as previously reported [10,11]. Therefore, red betacyanin distribution extends to cells beyond those with promoter activity but still specifically labels colonised tissues and is thus a promising candidate for the in vivo visualisation of AM symbiosis processes.

### Identification and validation of AM symbiosis marker genes in *Nicotiana benthamiana*

Transiently transformed *M. truncatula* hairy roots display varying degrees of transgene expression that could impact reporter intensity and functionality. To avoid this, we sought to establish betalains as reporters of AM symbiosis in stable transgenics. Here, we explored the classic model species *N. benthamiana*, which has been extensively used to develop methods of heterologous betalain production [23,31,43]. We first identified homologues of *MtPT4* (*NbPT5b*; Niben101scf02726g00004.1) and *MtBCP1* (*NbBCP1b*; Niben101Scf07438g04015.1) in *N. benthamiana* via BLASTP against the proteins from the

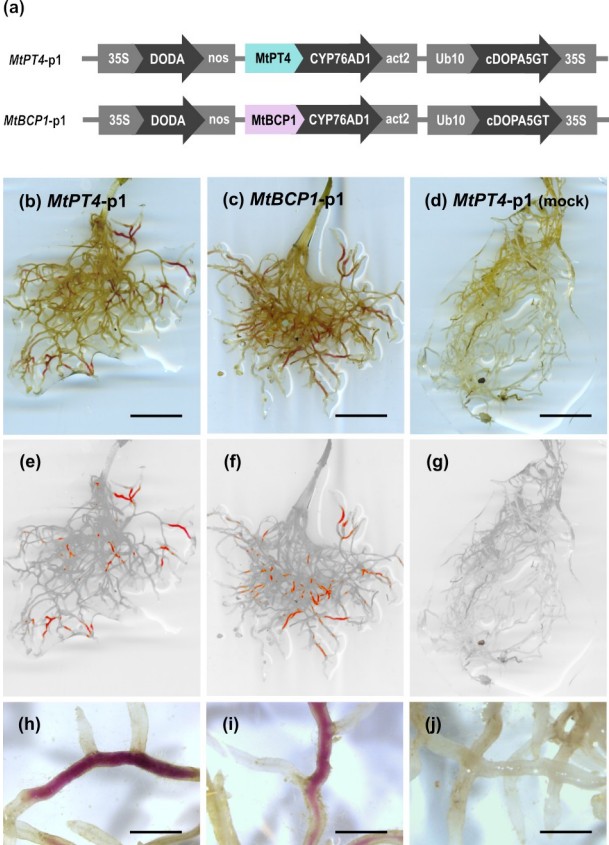

**Fig 2. Betalains can be produced in *Medicago truncatula* roots as a response to AM fungi colonisation.** (a) Schematic of the multigene vectors constructed for inducible betalain expression in *M. truncatula* roots where only the first gene of the betalain biosynthesis pathway is controlled by AM symbiosis–specific promoters. Expression of *MtPT4*-p1 (b, e, and h) and *MtBCP1*-p1 (c, f, and i) in roots of *M. truncatula* 4 weeks after inoculation with *Rhizophagus irregularis*. (d, g, and j) Example of *MtPT4*-p1 expressing root system mock inoculated with autoclaved *R. irregularis*. (e, f, g) Root system images filtered for red colouring only. Scale bar (b, c, and d), 1 cm; scale bar (h, i, and j), 1.5 mm. AM, arbuscular mycorrhiza.

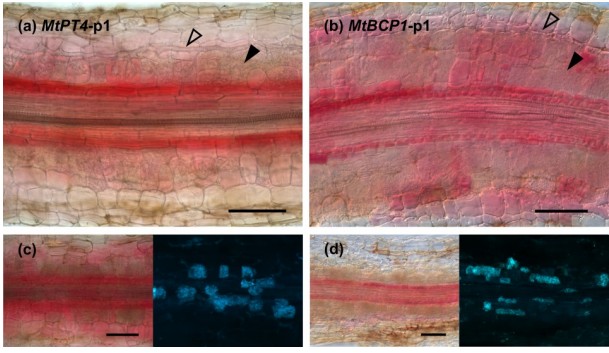

**Fig 3. Betalain accumulation is observed at different intensities in different tissue layers and appears higher in the endodermal cells adjacent to arbuscule-containing cortical cells.** Root sections of *Medicago truncatula* expressing *MtPT4*-p1 (a and c) and *MtBCP1*-p1 (b and d) 4 weeks after inoculation with *Rhizophagus irregularis*. (c and d) Left: Betacyanin pigments are visible in red; right: WGA-FITC staining of fungal structures in blue. Open arrows mark internal hyphae, and filled arrows signal cells containing arbuscules. Scale bar, 100 μm. WGA, wheat germ agglutinin.

most recent genome sequence (v1.01, Solgenomics) and confirmed the homology of the *N. benthamiana* to the *M. truncatula* loci, by phylogenetic analysis of the BCP and PT orthogroups (S3 Fig). To confirm inducibility of the *NbPT5b* and *NbBCP1b* genes during mycorrhizal symbiosis, we investigated their expression at different time points on colonised and non-colonised roots of *N. benthamiana* with *R. irregularis* (Fig 4A). We used the *R. irregularis* β-tubulin (*RiBTub*) gene as a fungal biomass marker to confirm successful colonisation. Expression of *NbPT5b* and *NbBCP1b* increased after 2 weeks and showed significantly elevated transcript levels 3 to 4 weeks postinoculation (Fig 4A). We therefore cloned promoter regions up to −1,068 bp and −1,231 bp upstream of the start of *NbPT5b* and *NbBCP1b* coding sequences, respectively (S1 Data). To address tissue specificity of *NbPT5b* and *NbBCP1b* promoters, we generated promoter–GUS fusions for expression in *N. benthamiana*. Stable *N. benthamiana* transformants expressing *NbPT5b-GUS* and *NbBCP1b-GUS* fusions displayed a GUS staining pattern specific to root areas colonised by *R. irregularis* (Fig 4B and 4C). In both cases, GUS staining was stronger in cells containing arbuscules but could also be observed in adjacent cells at varying intensities (Fig 4B and 4C). Non-colonised areas were consistently free of staining in *NbPT5b-GUS* and *NbBCP1b-GUS* roots. Altogether, the results support the use of *NbPT5b* and *NbBCP1b* promoters for the construction of a betalain-based AM symbiosis reporter system in *N. benthamiana*.

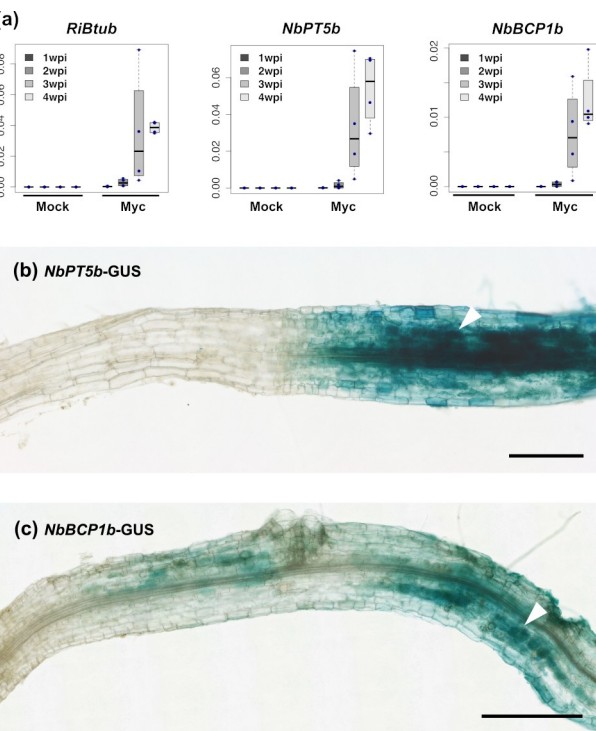

**Fig 4. Expression of *NbPT5b* and *NbBCP1b* genes is induced under colonisation with *Rhizophagus irregularis* in *Nicotiana benthamiana*.** (a) qRT-PCR analysis of *NbPT5b* and *NbBCP1b* shows that gene expression is induced 2 wpi with *R. irregularis* and increases with time and degree of colonisation. *Y* axes indicate expression levels relative to *N. benthamiana*'s elongation factor 1 alpha (NbEF). Expression of *RiBTub* is used as a fungal biomass marker for root colonisation. Raw qRT-PCR values can be found in S3 Data. (b and c) GUS staining of *NbPT5b*-GUS and *NbBCP1b*-GUS expressing *N. benthamiana* roots 4 weeks after inoculation with *R. irregularis*. GUS activity can only be observed in root areas colonised by *R. irregularis* and is more predominant in arbuscule-containing cells (white arrows). Scale bar, 500 μm. qRT-PCR, quantitative real-time polymerase chain reaction; RiBTub, *R. irregularis* β-tubulin; wpi, weeks postinoculation.

### Betalains visualise AM fungus colonisation in living *Nicotiana benthamiana* roots

We then generated multigene betalain reporter constructs for stable transformation in *N. benthamiana*. A first generation of vectors placed *CYP76AD1* under the control of the *NbPT5b* or *NbBCP1b* promoters, with *DODA* and *cDOPA5GT* driven by *35S* and *Ubi10* constitutive promoters, respectively. Hereafter, we refer to these multigene vectors with one AM fungal colonisation-specific promoter as *NbPT5b*-p1 and *NbBCP1b*-p1 (Fig 5A). We generated 16 independent transgenic *N. benthamiana NbPT5b*-p1 lines and 12 independent *NbBCP1b*-p1 lines. *NbPT5b*-p1 and *NbBCP1b*-p1 plants produced betalains in the roots upon colonisation with *R. irregularis* (Fig 5B and 5C). Betalain production was not visible in mock conditions for *NbPT5b*-p1 or *NbBCP1b*-p1 plants inoculated with autoclaved *R. irregularis* (S4 Fig). Dissection and microscopy of pigmented and nonpigmented root fragments revealed betalain presence colocalised with fungal colonisation structures (Fig 5D–5G).

To assess the extent to which red pigment co-occurs with intraradical fungal structures, we divided *NbPT5b*-p1 and *NbBCP1b*-p1 T1 plant root systems into red-pigmented and nonpigmented fragments and ink stained them separately for the presence of fungal structures (S5 Fig). We then imaged the stained root fragments and divided each root image into sections of similar length for the scoring of fungal presence. For both *NbPT5b*-p1 and *NbBCP1b*-p1 root systems, we found that all of the betacyanin-positive root fragments contained fungal structures and were extensively colonised (Table 1). White root segments of the *NbPT5b* line were

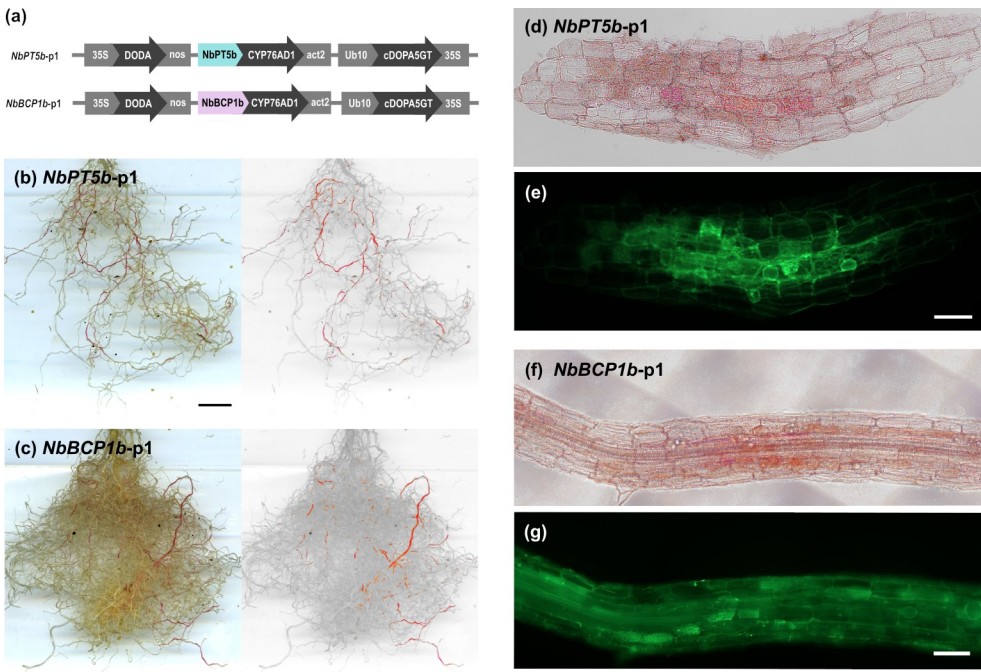

**Fig 5. Betalains can be produced in *Nicotiana benthamiana* roots as a response to AM fungi colonisation.** (a) Schematic of the multigene vectors constructed for inducible betalain expression in *N. benthamiana* roots where only the first gene of the betalain biosynthesis pathway is controlled by AM symbiosis–specific promoters. Expression of *NbPT5b*-p1 (b) and *NbBCP1b*-p1 (c) in whole root systems of *N. benthamiana* T1 plants 4 weeks after inoculation with *Rhizophagus irregularis*. Right: Root images are filtered for red colour. (d–g) Confocal microscopy of red *NbPT5b*-p1 (d and e) and *NbBCP1b*-p1 (f and g) transgenic root sections which were cut and stained with WGA-FITC to visualise fungal structures. (d and f) colour-filtered for red; (e and g) FITC green fluorescence. Scale bar (b and c), 1 cm; scale bar (d–g), 100 μm. AM, arbuscular mycorrhiza; WGA, wheat germ agglutinin.

**Table 1. Quantification of fungal structures observed in roots of *Nicotiana benthamiana* expressing *NbPT5b*-p1 and *NbBCP1*-p1 4 wpi with *Rhizophagus irregularis*.**

| Genotype | Pigmented | N | Percent of colonised root fragments (%) | | | Average length of root fragments colonised (%) | | |
|---|---|---|---|---|---|---|---|---|
| | | | IH | A | V | %IHC | %AC | %VC |
| *NbPT5b*-p1 | + | 45 | 100.0 | 100.0 | 97.8 | 77.9 ± 3.5 | 74.7 ± 3.9 | 62.3 ± 4.1 |
| | - | 35 | 22.9 | 17.1 | 14.3 | 4.8 ± 2.2 | 3.0 ± 1.4 | 1.1 ± 0.5 |
| *NbBCP1b*-p1 | + | 31 | 100.0 | 100.0 | 100.0 | 93.1 ± 2.4 | 86.6 ± 3.4 | 78.5 ± 4.0 |
| | - | 34 | 88.2 | 70.6 | 67.6 | 46.4 ± 5.7 | 33.6 ± 5.2 | 21.4 ± 4.3 |

We cut and divided roots into pigmented (red rows) and nonpigmented (noncoloured rows) fragments for ink staining. *N* refers to the total number of root fragments analysed for each condition. Table shows the number of root fragments containing fungal structures and the average extent of colonisation by these structures over the length of the root fragment. Error is shown as standard error.

%AC, percentage of arbuscule colonisation; %IHC, percentage of internal hyphae colonisation; %VC, percentage of vesicle colonisation; A, arbuscules; IH, internal hyphae; V, vesicles; wpi, weeks postinoculation.

overall much less likely to harbour fungal structures (max. 22.9% intraradical hyphae) compared to *NbBCP1b* (88.2%). Furthermore, the extent of fungal structures within individual white root segments was also much lower in *NbPT5b* than in *NbBCP1b* roots (4.8% versus 46.4%)(Table 1). This suggests that *NbPT5b*-driven *CYP76AD1* more reliably labels root segments containing fungal structures. Red root sections predominantly displayed colonisation by arbuscules and hyphae, and, to a lesser extent, vesicles, suggesting that accumulation of pigment in *N. benthamiana* is also associated with fully developed AM symbiosis but may decline at late stages (Table 1).

## Stable expression of *NbPT5b-p1* and *NbBCP1b-p1* can cause shoot developmental defects in *N. benthamiana*

Surprisingly, many *NbPT5b*-p1 and *NbBCP1b*-p1 transgenic lines displayed vegetative developmental defects of varying severity (S6 Fig). Specifically, 13 out of 16 *NbPT5b*-p1 and 11 out of 12 *NbBCP1*-p1 lines were affected with moderate to severe defects. Aberrant phenotypes were not visible in early stages of development but became prominent during maturation and flowering. Affected plants were smaller and displayed altered leaf morphology, ranging from curvy leaf edges to acute deformation of leaf shape and thickness (S6 Fig). We also observed defects in flower morphology, including stunted or absent perianth, and, in more severe cases, missing floral reproductive organs (S6 Fig). Only plants with mild to no phenotypic alterations were able to set seed, although seed set was not abundant. To address whether these phenotypes were linked to the expression of constitutive or AM-specific promoter–driven betalain genes, we performed semiquantitative reverse transcription PCR (RT-PCR) on abnormal leaves of *NbPT5b*-p1 transgenic plants. As expected, we detected *35S* promoter-driven *DODA* expression and *cDOPA5GT* in all analysed T0 lines, but, unexpectedly, *DODA* expression was increased in severely affected lines (S6 Fig). Surprisingly, we detected expression of AM-responsive *PT5b* promoter-driven *CYP76AD1* in T0 lines with light and moderate developmental defects (S6 Fig). Expression intensity of *CYP76AD1* appeared to decrease with phenotype severity and was barely detectable in lines displaying acute developmental defects. T1 seedlings from various *NbPT5b*-p1 transformant lines developed smaller leaves and shorter roots than wild-type (WT) seedlings (S7 Fig). T1 plants were nevertheless able to grow to adult stage, flower, and set seed. In addition to developmental defects, we also detected unanticipated accumulation of betanin by HPLC in leaves of T1 plants that descended from *NbPT5b*-p1 lines with mild to no phenotypic alterations (S8 Fig). Leaf betanin accumulation arose in

both inoculated and mock conditions (S8 Fig). In summary, we found that expression constructs with 2 constitutive biosynthesis genes frequently resulted in both developmental aberrations and unexpected aerial betalain expression.

## Expression of all betalain synthesis genes under AM symbiosis–specific promoters ameliorates defective phenotypes in *N. benthamiana*

To avoid the developmental defects associated with the constitutive expression of betalain genes in *N. benthamiana*, we sought to express all 3 betalain biosynthesis genes under AM symbiosis–specific promoters. Here, a second generation of vectors placed *CYP76AD1*, *DODA*, and *cDOPA5GT* under the control of either the *NbPT5b* or *NbBCP1* promoters. Hereafter, we refer to these multigene vectors as *NbPT5b*-p3 and *NbBCP1b*-p3 (Fig 6A). To test their performance, we generated 11 independent transgenic *NbPT5b*-p3 lines and 7 *NbBCP1b*-p3 lines and grew them with *R. irregularis* spores in a rhizotron setup. Root system pigmentation patterns of the *NbPT5b*-p3 and *NbBCP1b*-p3 reporter constructs performed similarly to *NbPT5b*-p1 and *NbBCP1b*-p1 (S9 Fig). Colonisation extent and fungal structures frequency in inoculated root systems did not differ between WT and *NbPT5b*-p3 and *NbBCP1b*-p3 reporter

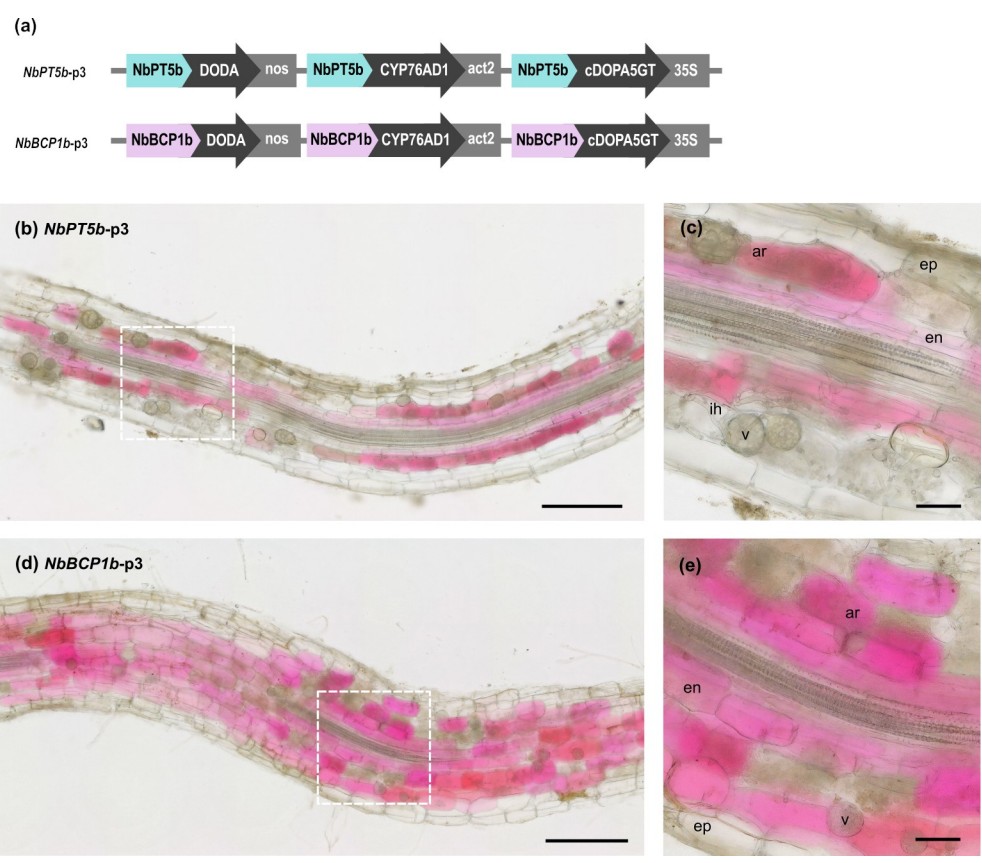

**Fig 6. Betalain production in *Nicotiana benthamiana* roots as a response to AM fungi colonisation via expression of the entire betalain pathway under AM symbiosis–specific promoters.** (a) Schematic of the multigene vectors constructed for inducible betacyanin expression in *N. benthamiana* roots with the 3 betalain biosynthetic genes controlled by the *NbPT5b* or the *NbBCP1* promoters. Expression of *NbPT5b*-p3 (b and c) and *NbBCP1b*-p3 (d and e) in roots of *N. benthamiana* 4 weeks after inoculation with *Rhizophagus irregularis*. Betacyanin production was not visible in mock conditions for any *NbPT5b*-p3 or *NbBCP1b*-p3 roots. (c and e) are a magnification of the areas delimited by the dashed squares. Scale bar (b and d), 500 μm; scale bar (c and e), 100 μm. AM, arbuscular mycorrhiza; Ar, arbuscules; en, endodermis; ep, epidermis; ih, internal hyphae; v, vesicle.

lines (S10 Fig). Once again, plants grown in mock conditions did not harbour any observable colouration despite being grown in non-sterile substrate, further suggesting that reporter expression is not activated by the presence of other microorganisms. This is also supported by the absence of red pigmentation upon inoculation of reporter lines with spores of the oomycete root pathogen *Phytophthora palmivora* (S11 Fig).

Dissection and microscopy of pigmented and nonpigmented *NbPT5b*-p3 and *NbBCP1b*-p3 root fragments inoculated with *R. irregularis* revealed once more that betalain presence colocalised with fungal colonisation structures (Fig 6). Arbuscule formation could be observed in all cortex cell layers in *N. benthamiana*. In *NbPT5b*-p3 and *NbBCP1b*-p3 roots, betalain pigmentation was confined to single cells that were clearly distinguishable. Pigmentation was, however, not restricted to arbuscule-containing cells and could also be detected in adjacent cells with no apparent arbuscule structures including cells of the endodermis (Fig 6). Similarly, some arbuscule-containing cells in the proximity of red pigmented cells appeared unpigmented both in *NbPT5b*-p3 and *NbBCP1b*-p3 roots (Fig 6). Importantly, only 1 out of the 11 lines in the *NbPT5b*-p3 genotype and 2 out of the 7 lines in the *NbBCP1b*-p3 genotype displayed the developmental defects seen with *NbPT5b*-p1 and *NbBCP1b*-p1 reporter constructs. Finally, the T1 generations derived from *NbPT5b*-p3 and *NbBCP1b*-p3 lines did not exhibit any discernible pigmentation in shoot tissues. Therefore, the expression of all betalain biosynthetic pathway genes under AM-specific promoters ameliorates the incidence of developmental alterations and nonspecific betalain production in *N. benthamiana* plants.

We also confirmed the functionality of constructs where all betalain synthesis genes are driven by arbusculated cell specific promoters in *M. truncatula*. We used the *MtPT4* promoter to drive the expression of betalain genes and included a constitutively expressed DsRed marker to facilitate selection of transformed hairy roots. Betalain colouration was produced in colonised roots and was associated with arbuscule presence (S12 Fig). However, once again, staining was not restricted to arbuscule-containing cells of the inner cortex and could be mainly observed in internal tissue layers like the endodermis, pericycle, and stele (S12 Fig). *M. truncatula Doesn't Make Infections 3* (*dmi3*) mutants fail to progress in intraradical AM fungal colonisation and are unable to engage in symbiosis [44,45]. Therefore, reporter expression in *dmi3* mutants should not harbour any colouration regardless of inoculum presence. Notably, red pigmentation was not observed in composite roots of *M. truncatula dmi3* mutants (S13 Fig). The efficiency of betalain accumulation as a marker of AM colonisation with *MtPT4* was similar to that observed for *NbPT5b* in *N. benthamiana* plants (S14 Fig and S1 Table).

## Promoter-controlled betalain biosynthesis allows for dynamic tracing of root colonisation processes

Expression of the betalain biosynthesis genes under the control of *NbPT5b* and *NbBCP1b* promoters in *N. benthamiana* also allowed for the noninvasive imaging of AM fungal root colonisation in root systems. We grew *NbPT5b*-p3 and *NbBCP1b*-p3 transgenic *N. benthamiana* lines in a vertical rhizotron-based setup with *R. irregularis* spores that maintained root system architecture by separating roots from the substrate with a porous black cloth. This also allowed for nondisruptive imaging on a flatbed scanner and subsequent image analysis using a selective colour filter for red pigment. At 52 days postinoculation (dpi), plants had developed significant root stretches with red pigmentation in both *NbPT5b*-p3 and *NbBCP1b*-p3 lines that was easily discriminated by computer imaging (Fig 7).

To assess AM colonisation dynamics over time, we also acquired consecutive images of intact *NbBCP1b*-p3 and *NbPT5b*-p3 rhizotrons at different days. This revealed increasing red pigment development within the root systems (Figs 8 and S15). Betalain pigmentation served

**Fig 7. Red pigment distribution in root systems of *NbBCP1b*-p3 and *NbPT5b*-p3 *Nicotiana benthamiana* plants colonised by *Rhizophagus irregularis*.** Images were taken at 52 dpi. (a and c) are reflective light images, and (b, d, and e) are filtered for red colouring only. (e) High magnification image of (d) showing varying red colouration in parallel running roots. Scale bar = 1.3 cm. dpi, days postinoculation.

as a reliable and long-lasting indicator of AM colonisation, as we did not observe root stretches that displayed red pigmentation and later lost it. Together, these data demonstrate that red pigmentation effectively traces fungal colonisation over time allowing for the dynamic and noninvasive assessment of root sections where the AM-responsive promoters have been activated.

## Discussion

AM symbiosis is a fundamental and widespread trait in plants that greatly expands the root surface area for nutrient uptake. AM symbiosis is consequently a key agronomic trait in the drive to enhance future crop yields through environmentally sustainable mechanisms. However, enhancing AM fungal association in crop species requires the tools to understand its fundamental dynamics across varying agronomic and ecological contexts. To this end, we demonstrate the use of betalain pigments as in vivo visual markers for the occurrence and distribution of AM fungal colonisation in the roots of *M. truncatula* and *N. benthamiana*. We have generated multigene vectors in which AM colonisation specific plant promoters control the expression of core betalain synthesis enzymes in the production of betalain pigments. We show that AM-specific promoter-controlled betalain pigmentation is a powerful macroscopic tool to report and trace fungal colonisation in vivo along the root (Fig 9).

Macroscopically, betalain colouration was specifically limited to regions of the root colonised by *R. irregularis* in both *M. truncatula* and *N. benthamiana*, as we detected few to no false positives. Furthermore, no betalain pigmentation was detected in mutant lines of *M. truncatula* impaired in AM symbiosis. Quantification of fungal structures revealed that pigmented root fragments were extensively colonised, showing arbuscule-containing cells over the majority of the root length for all tested promoters. We also observed some fungal structures in a low percentage of nonpigmented root fragments. In these cases, arbuscule colonisation was restricted over the length of the analysed root fragments, with the exception of *NbBCP1b* expressing *N. benthamiana* roots that exhibited a greater degree of unpigmented yet colonised root fragments. Thus, reporter ability to document the totality of colonisation events depends on the promoter chosen for vector construction. It appears that the *NbBCP1b* promoter-containing constructs are not equally activated in all colonisation events, or perhaps pigment accumulation becomes evident only at certain stages of colonisation. Promoter sequences of *PT4* homologues such as *NbPT5b* were able to report the majority of colonisation events in the root

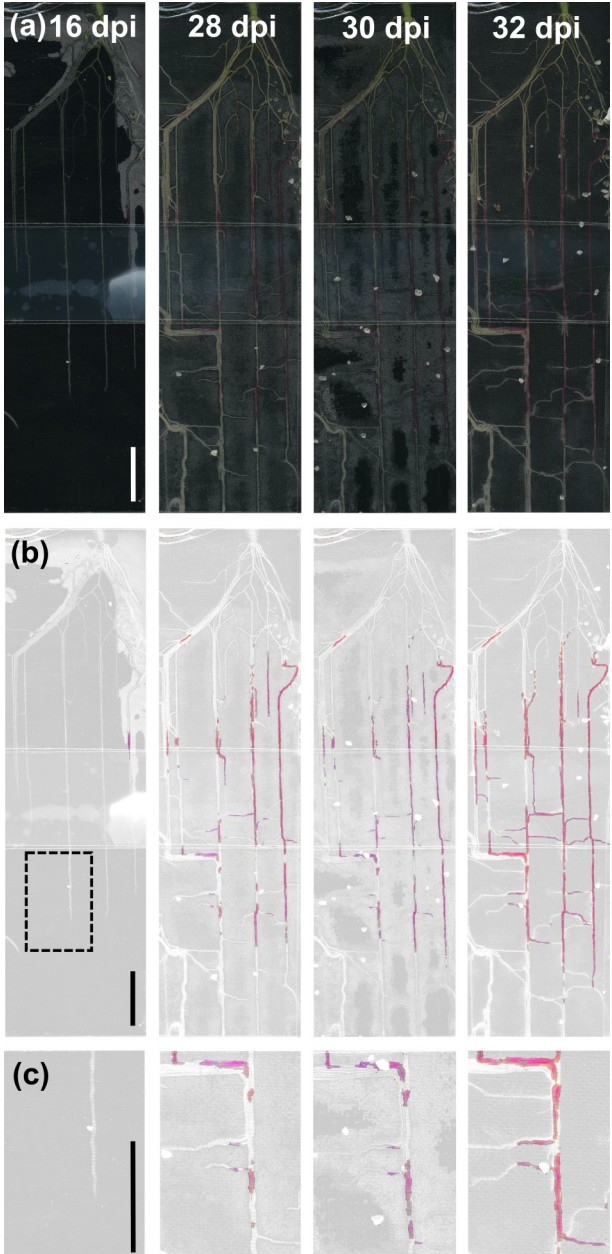

**Fig 8. *NbBCP1b* promoter-controlled betalain biosynthesis allows for dynamic tracing of root colonisation processes in *Nicotiana benthamiana*.** Transgenic *NbBCP1*-p3 plants grown in a rhizotron setup supplied with *Rhizophagus irregularis* spore inoculum imaged over time. (a) Reflective light images, (b) represent the same images filtered for red/magenta hues, and (c) is a magnification of the area delimited by the dashed square over time. Scale bar, 1 cm. dpi, days postinoculation.

system and are therefore preferred for reporting total root colonisation. A future solution to document total AM colonisation could involve the establishment of systems whereby the betalain biosynthesis genes are activated by transactivators, which could be then driven by promoters that are active at early, main, and late AM fungal colonisation stages [46–48]. Overall, betalain pigmentation effectively reports fungal colonisation with very little to no error and allows for simple selection of colonised root areas.

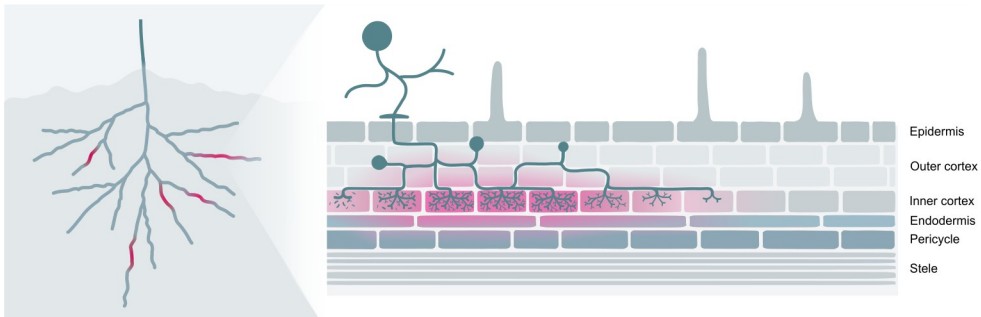

**Fig 9. Betalains can be used as markers for AM colonisation in plant roots.** Left: Red pigmentation is easily observable in whole plant root systems. Right: Red pigmentation is most prominent in colonised tissues as well as in adjacent tissue layers.

Microscopy of pigmented tissues revealed that betalain accumulation extended beyond arbusculated cells and into adjacent cell layers in both *M. truncatula* and *N. benthamiana*. In *M. truncatula*, betalain pigmentation was most prominent in the endodermis and pericycle cells adjoining arbusculated inner cortical cells (Fig 3). In *N. benthamiana*, we also observed betalain pigmentation in the endodermis and pericycle cells, yet pigmentation was more strongly retained in cortical cells (Fig 6). However, colouration in *N. benthamiana* was sometimes present in non-arbusculated cells adjacent to arbusculated cortical cells, consistent with previously observed GUS staining patterns. These extended patterns occur even when using constructs where all three biosynthesis enzymes are driven by AM-specific promoters and cannot therefore be attributed to constitutive gene expression artefacts. Such pigment accumulation in non-arbusculated cells may be the result of a former colonisation or of moderate degree of pigment migration. Betalains are water-soluble pigments and, in native betalain-pigmented species, are produced in the cytoplasm and then stored in vacuoles [49], although the mechanisms responsible for the intracellular transport of betalains are unknown. As small water-soluble compounds, betalains may have the potential to move symplastically through root plasmodesmata, although this remains unproven. Transient expression of betalains in leaves of *N. benthamiana* leads to macroscopically well delimited pigmented areas [23,31,43], but cellular migration across the boundary tissue has not been studied. Unpigmented cells harbouring arbuscules could also be a result of mechanical disruption during the sectioning of root tissues for microscopy, where cells that previously contained betalains lose pigmentation after being sliced open. Nevertheless, the betalain reporter system remains a highly effective marker of AM colonisation, especially at the macroscopic level.

The betalain biosynthetic pathway has been previously constitutively expressed in several plant species, including plants of the Solanaceae, with no report of observable developmental defects [31,40]. Yet, in our study, expression of betalain producing constructs where only *CYP76AD1* was under the control of AM-specific promoters led to developmental defects in *N. benthamiana* lines. Affected plants displayed dwarf phenotypes, altered leaf, and flower morphology and were unable to set seed (S6 Fig). Our findings could be explained by a mechanism whereby constitutive expression of either, or both, *DODA* and *cDOPA5GT* in absence of *CYP76AD1* expression cause developmental abnormality. Three observations are consistent with this hypothesis: (1) expression of constructs containing all three biosynthesis enzymes under symbiosis specific promoters substantially decreased the number of affected plants; (2) *DODA* expression appeared enhanced in leaves of severely affected plants; and (3) expression of *CYP76AD1* was detected in leaves of plants with mild defects and seemed to decrease with

phenotype severity. We speculate that, in absence of L-DOPA, the DODA enzyme could be promiscuously acting on alternative *N. benthamiana* substrates, with negative developmental consequences. This effect is alleviated by the presence of CYP76AD1, which provides the DODA enzyme with an abundance of its native substrate, L-DOPA, resulting in the production of inert betalain pigmentation. Constitutive *DODA* expression may therefore create a shoot developmental conflict that can be partially mitigated by compensatory expression of *CYP76AD1*. Our selection process could therefore have been biased towards T0 plants with a degree of escaped *CYP76AD1* expression, which would also explain the presence of betanin in a number of T1 plants descending from *CYP76AD1* shoot-expressing lines (S9 Fig). Further experimentation is required to support this hypothesis, but in any case, developmental defects and vegetative betanin expression can be avoided when all three biosynthesis enzymes are driven by AM symbiosis–specific promoters.

The most powerful application of our betalain-based AM reporter lies in its ability to non-invasively document colonisation in fully developed root systems over time (Figs 8 and S15). This will facilitate answering important questions in the field of AM symbiosis and its potential for agronomic improvement. These include understanding the dynamics of root system colonisation, including the time difference between lateral root emergence and its colonisation, or the differential colonisation susceptibility of different root orders [50]. Colonisation can be assessed in scenarios where plants compete with each other [51] or with shoot pests [52], different nutrient regimes, CO2 concentration, temperatures, soil structures [53], and other abiotic factors. A second important application is in the survey of induced plant genetic variation or fungal natural genetic variation that impacts on root system colonisation. Here, its use is only limited by the transformability of the plant species, and an important future step will be to test our approach in monocot crops, some of which have already been engineered to express betalain pigments in grains [41]. Finally, a third application derives from the ability to bulk collect betalain-pigmented root sections produced under AM fungal colonisation, especially when there is very little overall colonisation as is common in some symbiosis signalling pathway mutants or when colonisation is restricted to specific roots. Here, stage-specific early or late promoters will unlock targeted transcriptomic, proteomic, or metabolomic analysis of colonised roots without the dilution effect of non-colonised tissues. Such enrichment strategies may, for example, help in understanding communication mechanisms between AM fungi and plants. Targeted sampling of pigmented roots will also simplify low-throughput microscopy such as cryo-electron microscopy and drastically improve the signal-to-noise ratio inherent in current sampling processes. Selective sampling at red-to-white transition zones may additionally aid time-lapse microscopy of expanding fungal colonisation arrays within roots.

In summary, betalains are plant pigments with a strong potential as visual markers for the study of physiological and developmental processes in plants and microorganisms. Here, we have expanded the application of these versatile pigments to add to an ever increasing range of betalain-based technologies as reporters of AM fungi colonisation in plant roots. MycoRed complements currently used fungal visualisation techniques and constitutes a powerful tool that will be of great value for the plant–microbe research community to advance knowledge in the field of AM symbiosis.

## Materials and methods

### Plant material and growth conditions

*M. truncatula* plants were grown in dried industrial sand at 21˚C day temperature, 19˚C night temperature, 65% humidity, and a 16-hour photoperiod at 350 $\mu$moles·m$^{-2}$·s$^{-1}$ light intensity. *N. benthamiana* plants belong to a standard laboratory line maintained by selfing. *N.*

*benthamiana* plants were grown in F2 soil (Levington, Frimley, United Kingdom) in controlled greenhouse facilities maintained at 25˚C day temperature, 15˚C night temperature, ambient humidity, and a 16-hour photoperiod with 125 w·m$^{-2}$ of supplementary light. Watering was performed via dripping on capillary mats for 2 to 4 minutes every 6 hours.

Seed material for *N. benthamiana* MycoRed reporter lines (*NbPT5b*-p3 and *NbBCP1b*-p3) are available for shipping. Requests can be sent to corresponding author Dr. Sebastian Schornack.

## Plant inoculation with *R. irregularis*

*M. truncatula* and *N. benthamiana* plants were transferred to sand and inoculated with a commercial *R. irregularis* crude inoculum (consisting of oil-dry substrate, pieces of colonised maize roots, and *R. irregularis* spores and external hyphae) at a concentration of 1:20 (v/v) inoculum:sand. Plants were watered 3 times per week with 1x Long-Ashton solution (20x concentrate: potassium nitrate, 40 mM; calcium nitrate, 40 mM; monosodium phosphate, 15 mM; magnesium sulfate, 30 mM; manganese sulfate, 0.1 mM; copper sulfate, 0.01 mM; zinc sulfate, 0.01 mM; boric acid, 0.5 mM; sodium chloride, 1 mM; ammonium heptamolybdate, 0.7 μM; ethylenediaminetetraacetic acid ferric sodium salt, 0.6 mM).

## RNA extraction and gene expression analysis

*N. benthamiana* 2-week-old seedlings were inoculated with *R. irregularis*, and roots were harvested once a week for 4 weeks after inoculation. *M. truncatula MtPT4*-p1 hairy roots were sampled 4 weeks after inoculation. Mock control plants were inoculated with autoclaved *R. irregularis*. Total RNA from roots was extracted using the Qiagen RNeasy Mini Kit according to manufacturer's instructions (QIAGEN, Hilden, Germany). RNA quantity and quality were assessed by NanoDrop (Thermo Fisher Scientific, Waltham, Massachusetts, United States of America) and agarose gel electrophoresis. cDNA libraries were prepared using the iScript cDNA Synthesis Kit (Bio-Rad, Watford, UK) according to manufacturer's recommendations. Quantitative real-time PCR was performed on 1:20 diluted cDNA in combination with the Roche LightCycler 480 SYBR Green I Master Mix (Roche, Basel, Switzerland) and the oligonucleotide primers listed in S2 Table. Three technical replicates were performed per sample. Transcript levels from *N. benthamiana* were normalised using *N. benthamiana*'s elongation factor 1 alpha (*NbEF;* Niben101Scf04639g06007.1) and the ΔCt method as previously described [54]. Transcript levels from *M. truncatula* hairy roots were normalised using *M. truncatula*'s phosphatidylinositol 3- and 4-kinase gene belonging to the ubiquitin family (*MtUBQ*; Medtr3g091400) [55]. Expression of *R. irregularis*' β-tubulin (*RiBTub*; XM_025314309.1) and elongation factor 1 alpha (*RiEF*; XM_025321412.1) were used as fungal biomass markers for root colonisation.

## Promoter identification and isolation

Homologue colonisation marker genes in *N. benthamiana* were identified by BLASTP on NCBI with standard parameters using orthologous *M. truncatula* loci as bait. To confirm the homology of *BCP1* and *PT* loci in *N. benthamiana* and *M. trucatula*, orthogroups for *BCP* and *PT* gene families were retrieved from phytozome, aligned by codon using MAFFT, and analysed by FastTree. Trees were post-processed to include FastTree support values and to mask monophyletic tips. Leaf tissue from *N. benthamiana* and *M. truncatula* A17 plants was snap frozen in liquid nitrogen and stored at −80˚C. Frozen tissue was ground to a fine powder using 5-mm glass beads and a Tissue Lyser II homogeniser (QIAGEN). Genomic DNA extraction was performed on up to 100-mg ground tissue using the QIAGEN DNeasy Plant Mini Kit

according to the manufacturer's specifications. Resulting genomic DNA was used to amplify 5′ flanking regions for *M. truncatula*'s *PT4* and *BCP1* gene and *N. benthamiana*'s *PT5b* and *BCP1b* genes by PCR using Phusion High-Fidelity DNA polymerase (Thermo Fisher Scientific), with the oligonucleotide primers listed in S2 Table. Primers were designed to include a region of 836 bp upstream the start codon for *MtPT4*, 1,108 bp for *MtBCP1*, 1,068 bp for *NbPT5b* (Niben101Scf02726g00004.1), and 1,231 bp for *NbBCP1b* (Niben101Scf07438g04015.1). Amplification length for *N. benthamiana* promoter sequences was based on published promoter sizes in *M. truncatula* and suitable oligo binding sites [10,11]. PCR products were ligated into the pBlueScript SK cloning vector using T4 DNA ligase (New England Biolabs, Hitchin, UK), and plasmids were sequenced to confirm gene identity. Cloned sequences can be found in S1 Data.

## Generation of vectors for plant transformation

The construction of multigene vectors containing the betalain biosynthetic genes under the control of AM-specific promoters was carried out using Golden Gate cloning [56]. Cloning components were acquired from the MoClo Tool Kit [57,58] and the MoClo Plant Parts Kit [59] (Addgene) except for the 1,327 bp *A. thaliana* Ubiquitin 10 promoter (*Ubi10*), which was provided by Dr Nicola Patron (Earlham Institute, Norwich, UK). Isolated *MtPT4*, *MtBCP1*, *NbPT5b*, and *NbBCP1b* promoter regions were cloned into the pICH41295 level 0 accepting vector for promoter + 5′ UTR modules. Cloning proceeded through level 0, level 1, and level 2 modules using the one-pot, one-step type IIS restriction–mediated cloning reaction. GUS reporter vectors were constructed with an eGFP–GUS fusion gene placed under the control of the cloned arbuscular-responsive specific promoters and also included a Basta resistance gene under the *nos* promoter and terminator. Previously generated plasmids containing the betalain biosynthetic genes were used for the construction of betacyanin producing multigene vectors [23]. *DODAα1* and *CYP76AD1* sequences were originally isolated from *B. vulgaris* cDNA libraries, whereas the *cDOPA5GT* sequence was obtained from flowers of *Mirabilis jalapa*. Gene sequences had been previously domesticated to remove *Bsa*I and *Bpi*I restriction enzyme sites considering codon usage in *N. benthamiana*. The first set of multigene vectors created for hairy root transformation of *M. truncatula* and stable transformation of *N. benthamiana* contained *BvCYP76AD1* under the control of the arbuscular-responsive specific promoters (*MtPT4*, *MtBCP1*, *NbPT5b*, or *NbBCP1b*), *BvDODAα1* and *MjcDOPA5GT* under constitutive promoters, and included a Basta resistance gene under the *nos* promoter and terminator. We refer to these multigene vectors as *MtPT4*-p1, *MtBCP1*-p1, *NbPT5b*-p1, and *NbBCP1b*-p1, respectively (Figs 2A and 5A). The second set of multigene vectors created for root transformation of *M. truncatul*a and stable transformation of *N. benthamiana* contained *BvDODAα1*, *BvCYP76AD1*, and *MjcDOPA5GT* genes all under the control of the arbuscular-responsive specific promoters (*MtPT4*, *NbPT5b*, and *NbBCP1b*) and included a DSRed fluorescent marker in the *M. truncatula* vector or a Basta resistance gene in *N. benthamiana* vectors both under the *nos* promoter and terminator. We refer to these multigene vectors as *MtPT4*-p3, *NbPT5b*-p3, and *NbBCP1b*-p3, respectively (Figs S12A and 6A). Level 0 and level 1 vectors were confirmed by Sanger sequencing of the full inserts. Level 2 vectors were confirmed by Sanger sequencing of insert boundaries and by diagnostic digestion with restriction enzymes. MycoRed reporter vectors and level 0 plasmids used for their construction are listed in S3 Table. All vectors are available through Addgene (Cambridge, Massachusetts, USA).

## Hairy root transformation of *M. truncatula*

Hairy root transformation of *M. truncatula* was performed according to the previously described *Agrobacterium rhizogenes*–mediated method with modifications [60]. Multigene constructs were transformed into the *A. rhizogenes* ARQUA1 strain and grown at 28˚C in LB media supplemented with antibiotics (carbenicillin 12.5 mg/L, kanamycin 50 mg/L) until reaching an $OD_{600}$ of approximately 1.5. Cultures were then brought to a final $OD_{600}$ of approximately 1 in sterile distilled water. *M. truncatula* seeds were sterilised with 96% sulphuric acid for 5 minutes and pure sodium hypochlorite for 5 minutes, both steps followed by 5 to 10 washes with sterile distilled water. After a 2-day stratification process at 4˚C, seeds were germinated upside down on square dishes containing wet whatman paper overnight in the dark at 20˚C. Hypocotyl tips were then cut, and seedlings were suspended in *A. rhizogenes* cultures with acetosyringone (0.2 μM). Seed coats were removed, and seedlings were moved to solid mycorrhisation media (potassium nitrate, 80 mg/L; magnesium sulfate, 731 mg/L; potassium chloride, 65 mg/L; monopotassium phosphate, 4.8 mg/L; calcium nitrate, 288 mg/L; manganese (II) chloride, 6 mg/L; boric acid, 1.5 mg/L; zinc sulfate, 2.65 mg/L; sodium molybdate, 0.0024 mg/L; copper sulfate, 0.13 mg/L; potassium iodide, 0.75 mg/L; ethylenediaminetetraacetic acid ferric sodium salt, 8 mg/L; glycine, 3 mg/L; myoinositol, 50 mg/L; nicotinic acid, 0.5 mg/L; pyridoxine hydrochloride, 0.1 mg/L; thiamine hydrochloride, 0.1 mg/L; phytagel, 4 g/L; and pH 5.5) to grow vertically for 7 days at 20˚C and indirect lighting. Newly formed roots were then sectioned in half, and seedlings were transferred to mycorrhisation media supplemented with cefotaxime (250 μg/mL) and covered with Whatman's filter paper. Plants were grown vertically for 4 weeks at 20˚C and indirect lighting to allow for transgenic root development. Plants were then transferred to sand and subjected to fungal colonisation with *R. irregularis*. For constructs containing DsRed, all non-transformed roots were cut off before acclimation and inoculation. DSRed fluorescence was detected with a Leica M165 FC fluorescence dissecting stereomicroscope (Leica Biosystems, Wetzlar, Germany) using a filter with excitation and emission wavelengths of 510 to 560 nm and 590 to 650 nm, respectively.

## Stable transformation of *N. benthamiana*

*N. benthamiana* stable transformation was performed according to the previously described *A. tumefaciens*–mediated method with modifications [61]. Fully developed leaves of 4-week-old *N. benthamiana* plants were agro-infiltrated with *A. tumefaciens* carrying the constructs of interest according to previously described methods [61]. All constructs were transformed into the *Agrobacterium tumefaciens* GV3101 strain and grown at 28˚C in LB media supplemented with antibiotics (gentamycin 25 mg/L, rifampicin 60 mg/L, kanamycin 50 mg/L) until reaching an $OD_{600}$ of approximately 1.5. Cultures were then brought to a final $OD_{600}$ of approximately 0.3 in infiltration media (10 mM MgCl2, 0.2 mM acetosyringone, 10 mM MES at pH 5.6). Infiltration was performed on the abaxial surface of young expanding leaves of 4-week-old *N. benthamiana* plants. Three days after infiltration, full leaves were excised and cut into 1 to 2 cm$^2$ leaf squares. Leaf explants were sterilised in 70% ethanol for 5 minutes, 20% sodium hypochlorite supplemented with Tween 20 (0.1%) for 20 minutes, and washed thoroughly with sterile water. Sterilised leaf explants were then transferred adaxial side up to fresh selection media (1X Murashige and Skoog basal salt mixture, 1X Gamborg's B5 vitamins, 1% sucrose, 0.59 g/L MES, 2.0 mg/L BAP, 0.05 mg/L NAA, 0.4% Agargel, pH 5.7) supplemented with cefotaxime (500 mg/L), timentin (320 mg/L), and phosphinothricin (2 mg/L). Explants were subcultured onto new fresh media weekly until the appearance of first shoots, which were excised and planted in rooting media (½ Murashige and Skoog basal salt mixture, 0.5% sucrose, 0.25% Gelrite, 0.05mg/L NAA, pH 5.8) supplemented with augmentin (500 mg/L), timentin (320 mg/L),

and phosphinotricin (2 mg/L). Approximately after 2 weeks, transgenic plantlets with growing root systems were transferred to jars with sterile peat blocks. Transgenic plants were transplanted to the glasshouse when fully developed and allowed to grow until flowering and harvesting of seeds.

### RT-PCR gene expression analysis of *N. benthamiana* stable lines

*N. benthamiana* leaf tissue was snap frozen in liquid nitrogen and stored at −80˚C. Frozen tissue was ground to a fine powder using 5 mm glass beads and a Tissue Lyser II homogeniser (QIAGEN). RNA extraction was performed using the Concert Plant RNA Reagent (Invitrogen, Carlsbad, California, USA) followed by the TURBO DNA-free kit (Ambion, Carlsbad, California, USA) to remove DNA. RNA concentration was quantified by Nanodrop, and RNA integrity was assessed by agarose gel electrophoresis. cDNA libraries were prepared using BioScript Reverse Transcriptase (Bioline Reagents, London, UK) and oligo dT primers, according to manufacturer's recommendations. RT-PCR was performed on a 1:10 cDNA dilution, using the KAPA 2G Fast DNA polymerase kit (KAPA Biosystems, Wilmington, Massachusetts, USA) and the oligonucleotides specified in S2 Table.

### Histochemical staining for GUS activity

Transgenic *M. truncatula* and *N. benthamiana* roots carrying the *MtPT4*::*eGFP*:*GUS*, *MtBCP1*::*eGFP*:*GUS*, *NbPT5b1*::*eGFP*:*GUS*, and *NbBCP1b*::*eGFP*:*GUS* sequences were harvested 4 weeks after inoculation and treated with 80% acetone at −20˚C for 30 minutes. Roots were then washed with PBS for 5 minutes and incubated in a staining solution containing 100 mM sodium phosphate pH 7.0, 5 mM potassium ferrocyanide, 5 mM potassium ferricyanide, and 2 mM 5-bromo-4-chloro-3-indoxyl-β-D-glucuronidacid (X-gluc) for 3 hours at 37˚C. Roots were then washed twice with PBS and prepared for sectioning. *M. truncatula* roots were cast in paraffin wax, sectioned to a 5-μm width with a HM 340E rotary microtome (Thermo Fisher Scientific), and observed with an Olympus BX41 microscope (Olympus, Tokyo, Japan). *N. benthamiana* roots were embedded in 4% low melting point agarose, sectioned to a 100-μm width with a Leica VT1200S vibratome (Leica Biosystems) and imaged with a Keyence VHX-5000 microscope (Keyence, Osaka, Japan).

### Staining and quantification of fungal structures

Ink staining of fungal structures was performed in *M. truncatula* and *N. benthamiana* roots as previously described [15]. In brief, roots were incubated with 10% potassium hydroxide at 95˚C for 5 minutes and washed 3 times with sterile deionised water. Roots were then incubated in a solution of 5% pen ink (Sheaffer, Providence, Rhode Island, USA) and 5% acetic acid at 95˚C for 2 minutes, washed with 5% acetic acid and then 3 times with sterile deionised water. *N. benthamiana* roots were additionally incubated with ClearSee [62] for 20 seconds and washed 3 times with sterile deionised water. Stained root fragments were then mounted in parallel on a glass slide and imaged with a Keyence VHX-5000 digital microscope (Keyence). Each root image was divided into equally distant sections by digitally applying a 0.75 mm × 0.75 mm grid, and presence or absence of fungal structures (internal hyphae, arbuscules, and vesicles) were recorded in each square for each root. The length of root fragment colonised was calculated as the percentage of the number of squares containing fungal structures from the total number of squares in each root. Results were averaged for each genotype and condition. For wheat germ agglutinin-FITC conjugate (WGA-FITC) staining in *M. truncatula*, roots were incubated in 1 mg/mL of WGA-FITC conjugate (Sigma-Aldrich, Saint Louis, Missouri, USA) in potassium phosphate buffer 0.1 M (pH 7.2) 10 minutes in the dark, rinsed

several times in potassium phosphate buffer 0.1 M (pH 7.2), and mounted on glass slides. For staining in *N. benthamiana*, roots were incubated in WGA-FITC conjugate in potassium phosphate buffer 0.1 M (pH 7.2) supplemented with sucrose (3%) for 1 hour in the dark.

## Dissection and microscopy of roots for betalain visualisation

*M. truncatula* roots expressing *MtPT4*-p1, *MtBCP1*-p1, and *MtPT4*-p3 were harvested 4 weeks after inoculation. Roots were hand-sectioned and mounted in potassium phosphate buffer 0.1 M (pH 7.2) on glass slides. Sections were imaged using a Zeiss Axioimager microscope (Zeiss, Oberkochen, Germany) with brightfield and UV light. *N. benthamiana* roots expressing *NbPT5b*-p1 and *NbBCP1b*-p1 were harvested 4 weeks after inoculation, embedded in 4% agarose supplemented with sucrose (3%) and sectioned to a 50-μm width with a Leica VT1200S vibratome (Leica Biosystems). Sections were then mounted on a glass slide and stained with WGA-FITC conjugate as previously described. Root sections were visualised with a Leica TCS SP8 confocal laser scanning microscope equipped with a Leica DFC 7000T camera and HyD detectors (Leica Biosystems). *N. benthamiana* roots expressing *NbPT5b*-p3 and *NbBCP1b*-p3 were harvested 4 weeks after inoculation, embedded in 4% agarose, and sectioned to a 100-μm width with a Leica VT1200S vibratome (Leica Biosystems). Sections were then mounted on a glass slide and imaged with a VHX-7000 Keyence digital microscope (Keyence). Root images were loaded into GIMP 2.8.2. Red colouring was visualised by adjusting Hue/Saturation via Colors menu. A settings file is provided as S2 Data.

## Betalain extraction and detection using HPLC

Extraction and liquid chromatography of betalains in *N. benthamiana* leaf samples were performed as previously described [43]. In brief, leaf tissue samples were snap frozen in liquid nitrogen and grinded to a fine powder using mortar and pestle. Betalains were extracted overnight at 4˚C in 80% aqueous methanol with 50 mM ascorbic acid with a volume of 1 mL extraction buffer per 50 mg leaf tissue. After extraction, samples were clarified twice by centrifuging at 12,000 g for 5 minutes and collecting the supernatants. Liquid chromatography analysis was performed using a Thermo Fisher Scientific Accela HPLC autosampler (Thermo Fisher Scientific) and pump system incorporating a photodiode array detector. Betalains were separated using a Luna Omega column (100 Å, 5 μm, 4.6 × 150 mm) from Phenomenex (Torrance, California, USA) under the following conditions: 3 minutes, 0% B; 3 to 19 minutes, 0% to 75% B; and 7 minutes, 0% B where mobile phase A was 0.1% formic acid in 1% acetonitrile, and solvent B was 100% acetonitrile, and at a flow rate of 500 μL/min. The betacyanin betanin was quantified since it has been shown to be the predominant pigment arising from the combined expression of *DODA*, *CYP76AD1*, and *cDOPA5GT* [23]. Betanin was detected by UV/VIS absorbance at a wavelength of 540 nm. Identification and quantification of betanin were carried out using a commercially available *B. vulgaris* extract (Tokyo Chemical Industry UK, Oxford, UK).

## Rhizotron

The rhizotron consists of a 10 cm × 10 cm square petri dish where one of the sides of the base of the petri dish was removed to allow shoots to grow out. Ten-day-old *N. benthamiana* seedlings were placed on the base of the petri dish, covered with a coarse woven black cloth layer, followed by a layer of sand containing AM fungal spore inoculum. The lid was fixed using transparent adhesive tape. Rhizotrons were placed in black plastic bags and incubated in upright position over 52 days. For imaging, the plastic bag was removed, and rhizotrons were placed on a flatbed scanner. Images were loaded into GIMP 2.8.2. Red colouring was visualised by adjusting Hue/Saturation via Colors menu. A settings file is provided as S2 Data.

## *Phytophthora* infection assay

*N. benthamiana* seeds were sterilised for 5 minutes in 70% ethanol, followed by 5 minutes in 2% sodium hypochlorite solution, and washed 5 times with sterile water. The seeds were germinated on half-strength Murashige & Skoog medium including B5 vitamins (Duchefa Biochemie, Haarlem, the Netherlands) with 1% bactoagar, pH 5.7, and allowed to grow for 17 days at 25°C under a 16-hour photoperiod. Roots of 17-day-old seedlings were inoculated with 10 μL zoospore solution from *P. palmivora* LILI [63], which had been transformed anew with a pTOR-tdTomato fluorescent reporter provided by Dr Stephen Whisson (The James Hutton Institute, Dundee, UK) following the protocol described by Evangelisti and colleagues [64]. Zoospores were harvested as previously described [65] and diluted to the concentration of 20,000 zoospores/mL (200 zoospores/seedling). Control (mock) plants were inoculated with 10 μL of sterile water. Inoculated seedlings were grown at 25°C under a 24 hours photoperiod for 9 days. Inoculated seedlings were imaged using Leica M165FC microscope for visualising the extent of infection by *P. palmivora* and an Epson Perfection Flatbed Scanner (Epson UK, Hemel Hempstead, UK) using default settings and a resolution of 600 dots per inch for full-colour images to record betalain pigmentation.

## Supporting information

**S1 Fig. Expression of AM symbiosis marker genes *MtSTR* and *MtRAM2* in *Medicago truncatula MtPT4*-p1 composite roots is induced after inoculation with *Rhizophagus irregularis* and shows evidence of functional symbiosis.** Expression of *RiBTub* and *RiEF* is used as a fungal biomass marker for root colonisation. Gene expression was analysed by qRT-PCR 4 wpi. Mock roots were inoculated with autoclaved *R. irregularis* inoculum. Error bars represent standard errors of 3 biological replicates. Statistical *p*-value (*p*) was obtained via unpaired 2-sampled Student *t* test. Data underlying this figure can be found in S3 Data. AM, arbuscular mycorrhiza; qRT-PCR, quantitative real-time polymerase chain reaction; RiBTub, *R. irregularis* β-tubulin; RiEF, *R. irregularis* longation factor 1-alpha; wpi, weeks postinoculation.
(PDF)

**S2 Fig.** GUS staining of *Medicago truncatula* roots expressing *MtPT4*::*GUS* (a and b) and *MtBCP1*::*GUS* (c and d). GUS staining confirms promoter expression is limited to single cells in typically arbusculated tissue layers (inner cortex). Scale bar, 100 μm.
(PDF)

**S3 Fig.** Phylogenetic analysis of the PT and BCP1 orthogroups containing the (a) *MtPT4* and *NbPT5b* homologues and (b) *MtBCP1* and *NbBCP1b* homologues, respectively.
(PDF)

**S4 Fig.** Six-week old *Nicotiana benthamiana* T1 plants from *NbPT5b*-p1 (a–d) and *NbBCP1*-p1 (e–h) expressing lines. (a, b, e, and f) Plants 4 wpi with *Rhizophagus irregularis*. (c, d, g, and h) Plants descending from same lines mock inoculated with autoclaved *R. irregularis* inoculum. Scale bar, 1 cm. wpi, weeks after inoculation.
(PDF)

**S5 Fig.** Ink staining of *Nicotiana benthamiana* roots expressing *NbPT5b*-p1 (a–d) and *NbBCP1b*-p1 (e–h) after inoculation with *Rhizophagus irregularis*. (a, b, e, and f) Root fragments that displayed betalain colouration before ink staining are represented by a red "+" sign. (c, d, g, and h) Root fragments that displayed no colouration before ink staining are represented by a grey "−"sign. (b, d, f, and h) are amplified images of the area delimited by the

dashed squares. Scale bar, 1 mm.
(PDF)

**S6 Fig. Expression of *NbPT5b*-p1 in *Nicotiana benthamiana* stable transformants leads to developmental defects.** Figure shows examples of T0 plants that varied in the severity of observable developmental defects and were phenotypically classified as severe (S; a, d, and e), moderate (M; b, f, and g), and light (L; c). Only plants with a light phenotype were able to set seeds, although not abundantly. Defects affected overall plant size and leaf and flower morphology. Severely affected plants displayed thicker leaves with strong deformation commonly showing a bifurcated shape (d) and flowers lacking reproductive organs (e). Moderately affected plants displayed altered leaf elongation with curvy edges (f) and flowers with a shorter (or lacking) perianth (g). Light phenotypes included slight leaf deformation, and flowers with shorter perianth but typical overall plant growth (c). (h) Semiquantitative RT-PCR analysis of betalain biosynthetic transgene expression in leaves of *N. benthamiana* T0 stable *NbPT5b*-p1 transformants. Line 18 displayed light developmental defects and was able to set seed, line 6 displayed moderate developmental defects, and lines 1 and 15 displayed severe developmental defects. NbEF1α is used as housekeeping control. Positive control (+) is *NbPT5b*-p1 plasmid DNA. Negative control (−) is water. −*RT* stands for minus reverse transcriptase control generated from line 18. Original gel images can be found in S4 Data. RT-PCR, reverse transcription PCR.
(PDF)

**S7 Fig. One-week-old *Nicotiana benthamiana* T1 seedlings from *NbPT5b*-p1 and *NbBCP1b*-p1 expressing lines.** *NbPT5b*-p1 lines developed shorter roots and smaller leaves, which also appeared lightly pigmented in some cases. *NbPT5b*-p1 lines 18, 19, and 23, and the *NbBCP1b*-p1 line 32, were able to produce pigments upon colonisation with *R. irregularis*. *NbPT5b*-p1 line 23 did not produce any pigments upon colonisation.
(PDF)

**S8 Fig. Betanin detection by HPLC performed on T1 pigmented leaf tissue of *NbPT5b*-p1 expressing *Nicotiana benthamiana* lines.** (a) Some T1 plants show pigmentation in the leaves at varying degrees. (b) Liquid chromatography of pigmented leaves and controls. Pigmented leaves of *NbPT5b*-p1 expressing plants both in inoculated (myc) and non-inoculated (mock) conditions showed a peak at the expected retention time for betanin (red dotted line). Betanin standard is a commercially available *Beta vulgaris* extract and displays 2 peaks corresponding to the 2 betanin isomers commonly present in beet extracts. *NbPT5b*::*GUS* leaf tissue of same age T1 plants was used as negative control. Data underlying this figure can be found in S3 Data.
(PDF)

**S9 Fig.** Root systems of *NbPT5b*-p3 (a–d) and *NbBCP1b*-p3 (e–h) *Nicotiana benthamiana* plants colonised by *Rhizophagus irregularis*. Images were taken at 52 dpi. (a, b, e, and f) Plants inoculated with *R. irregularis*. (c, d, g, and h) Plants descending from the same lines mock inoculated with autoclaved inoculum. (a, c, e, and g) are reflective light images, and (b, d, f, and h) are filtered for red colouring only. Scale bar, 1.3 cm. dpi, days postinoculation.
(PDF)

**S10 Fig. Quantification of root AM colonisation shows no differences between *Nicotiana benthamiana* WT and reporter lines.** *N. benthamiana* plants were inoculated with *Rhizophagus irregularis* and ink stained 4 wpi. (a) Colonisation extent in *N. benthamiana* WT versus lines expressing *NbBCP1b*-p3 (BCP1-19 and BCP1-24) and *NbPT5b*-p3 (PT5b-16 and PT5b-

21). Colonisation extent is calculated as the percentage of root containing arbuscules, vesicles, or internal hyphae over the total root system. (b) Frequency of arbuscules and vesicles recorded in colonised roots of *N. benthamiana* WT versus lines expressing *NbBCP1b*-p3 and *NbPT5b*-p3. Frequency is calculated as the percentage of root containing arbuscules or vesicles over the extent of root colonised by *R. irregularis*. Individual points represent individual plants. Error bars represent standard errors. Data underlying this figure can be found in S3 Data. AM, arbuscular mycorrhiza; wpi, weeks after inoculation; WT, wild-type.
(PDF)

**S11 Fig. The oomycete root pathogen *Phytophthora palmivora* does not induce betalain pigment development in *NbPT5b*-p3 and *NbBCP1b*-p3 *Nicotiana benthamiana* lines within 9 dpi.** A total of 16 transgenic betalain reporter lines per construct were grown on pet-ridishes: 8 were not infected (mock), and 8 were infected with red fluorescent *P. palmivora* LILI-td zoospores. Plates were subsequently imaged under an epifluorescent microscope at 2 dpi (a) to document pathogen colonisation on roots and via flatbed scanning at 2 dpi (b) and 9 dpi (c). Application of a red filtering did not show any betalain pigment development. Scale bar, 50 mm. dpi, days postinfection.
(PDF)

**S12 Fig. Betacyanin production in *Medicago truncatula* roots as a response to AM fungi colonisation via expression of the entire betalain pathway under an AM symbiosis–specific promoter.** (a) Schematic of the multigene vector constructed for inducible betalain expression in *M. truncatula* roots with the 3 betalain biosynthetic genes controlled by the *MtPT4* promoter. (b–e) Expression of *MtPT4*-p3 in roots 4 wpi with *Rhizophagus irregularis* led to pigment accumulation in roots. (f–i) Mock inoculated roots expressing *MtPT4*-p3 did not show any pigment production. (b and f) Images taken under reflective light, and (c and g) are filtered for red colouring only. (d and h) Close up root images in bright field, and (e and i) and filtered to observe DSRed fluorescence. (j) Root section of *M. truncatula* expressing *MtPT4*-p3. Betalain accumulation correlates with arbuscule formation but can be observed mainly in the endodermal layer adjacent to arbuscule-containing cortical cells, pericycle, and steele. Open arrows mark internal hyphae and filled arrows signal cells containing arbuscules. Scale bar (b and f), 1 cm; scale bar (e and i), 1.5 mm; scale bar (j), 150 μm. AM, arbuscular mycorrhiza; wpi, weeks after inoculation.
(PDF)

**S13 Fig. Hairy roots of *dmi3* mutant *Medicago truncatula* plants expressing *MtPT4*-p3 do not produce any perceptible betalain colouration 4 wpi with *Rhizophagus irregularis*.** (a and b) Example of an A17 *M. truncatula* root system expressing *MtPT4*-p3 able to produce betalains upon inoculation. (c and d) Example of *dmi3 M. truncatula* root system expressing *MtPT4*-p3. (a and c) Images taken under reflective light, and (b and d) are filtered for red colouring only. Scale bar, 1 cm. wpi, weeks after inoculation.
(PDF)

**S14 Fig.** (a) Schematic of the fungal visualisation and quantification process of *Medicago truncatula* hairy roots after inoculation with *Rhizophagus irregularis*: (1) *M. truncatula* transgenic hairy roots express the DSRed marker and can be selected through detection of fluorescence under 510–560 nm; (2) DSRed positive roots are cut and divided into betalain producing and nonproducing root fragments; (3) pigmented and nonpigmented root fragments are ink stained separately (betalain colouration fades upon incubation with potassium hydroxide during the ink staining process); and (4) ink-stained root fragments are mounted in glass slides and imaged for visualisation and quantification of fungal structures. (b) Example of *MtPT4*-p3

pigmented root fragments showing fungal colonisation. (c) Example of *MtPT4*-p3 nonpigmented uncolonised root fragments. Scale bar, 100 μm.
(PDF)

**S15 Fig.** *NbPT5b* **promoter-controlled betalain biosynthesis allows for dynamic tracing of root colonisation processes in** *Nicotiana benthamiana*. Transgenic *NbPT5b*-p3 plants grown in a rhizotron setup supplied with *Rhizophagus irregularis* spore inoculum imaged over time. (a) Reflective light images, and (b) represent the same images filtered for red/magenta hues. Scale bar, 1 cm. dpi, days postinoculation.
(PDF)

**S1 Table. Quantification of fungal structures observed in hairy roots of** *Medicago truncatula* **expressing** *MtPT4*-p3 4 wpi with *Rhizophagus irregularis*. We selected roots for DSRed fluorescence and then cut and divided them in pigmented (red rows) and nonpigmented (noncoloured rows) fragments for ink staining. *N* refers to the total number of root fragments analysed for each condition. Table shows the number of root fragments containing fungal structures and the average extent of colonisation by these structures over the length of the root fragment. Error shown as standard error. %AC, percentage of arbuscule colonisation; %IHC, percentage of internal hyphae colonisation; %VC, percentage of vesicle colonsation; A, arbuscules; IH, internal hyphae; V, vesicles; wpi, weeks postinoculation.
(PDF)

**S2 Table. List of oligonucleotide primers used in this study.**
(PDF)

**S3 Table. List of vectors used in this study. Level 0 vectors used for construction of level 2 multigene vectors, which are also listed as MycoRed reporter constructs.** All vectors are available to obtain free of charge through Addgene. Plasmid maps and sequence files can be downloaded in the Addgene website with the catalogue numbers provided.
(PDF)

**S1 Data. AM-specific promoter sequences used for vector construction.** All sequences were selected immediately upstream the start codon of the gene.
(PDF)

**S2 Data. Hue/saturation settings for GIMP software used for filtering of betalain red colouration.**
(TXT)

**S3 Data. Raw data values for Figs 4A and S1, S8b and S10.**
(XLSX)

**S4 Data. Uncropped gel images used in S6 Fig.**
(PDF)

## Acknowledgments

We would like to thank Liron Shenhav for maintenance of *N. benthamiana* plants, Bo Xu and Edouard Evangelisti for help with wax casting, Nicola Patron for providing *Ubi10* promoter, Stephen Whisson for providing pTOR-tdTomato reporter, Edouard Evangelisti and Sabine Brumm for imaging, and Philip Carella for comments on the manuscript. *P. palmivora* research is covered by the Department for Environment, Food and Rural Affairs' (Defra) plant health license 114614/208745/4.

## Author Contributions

**Conceptualization:** Samuel F. Brockington, Sebastian Schornack.

**Data curation:** Alfonso Timoneda, Temur Yunusov, Samuel F. Brockington, Sebastian Schornack.

**Formal analysis:** Alfonso Timoneda, Clement Quan, Aleksandr Gavrin, Samuel F. Brockington.

**Funding acquisition:** Samuel F. Brockington, Sebastian Schornack.

**Investigation:** Alfonso Timoneda, Temur Yunusov, Clement Quan, Aleksandr Gavrin.

**Methodology:** Alfonso Timoneda, Temur Yunusov, Aleksandr Gavrin, Samuel F. Brockington, Sebastian Schornack.

**Project administration:** Samuel F. Brockington, Sebastian Schornack.

**Resources:** Samuel F. Brockington, Sebastian Schornack.

**Supervision:** Samuel F. Brockington, Sebastian Schornack.

**Visualization:** Alfonso Timoneda, Temur Yunusov, Aleksandr Gavrin, Sebastian Schornack.

**Writing – original draft:** Alfonso Timoneda.

**Writing – review & editing:** Alfonso Timoneda, Samuel F. Brockington, Sebastian Schornack.

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
