## [Editor Report · Decision Letter 0]

7 Aug 2020

Dear Dr Schornack, 

Thank you for submitting your manuscript entitled "MycoRed: Betalain pigments as in vivo real-time visual markers for arbuscular mycorrhizal colonisation of root systems" for consideration as a Methods and Resources by PLOS Biology.

Your manuscript has now been evaluated by the PLOS Biology editorial staff as well as by an academic editor with relevant expertise and I am writing to let you know that we would like to send your submission out for external peer review.

Please re-submit your manuscript within two working days, i.e. by Aug 11 2020 11:59PM.

Kind regards,

Aaron

Aaron Nicholas Bruns, Ph.D.,

Associate Editor

PLOS Biology

---

## [Decision Letter · Decision Letter 1]

29 Sep 2020

Dear Dr Schornack,

Thank you very much for submitting your manuscript "MycoRed: Betalain pigments as in vivo real-time visual markers for arbuscular mycorrhizal colonisation of root systems" for consideration as a Methods and Resources at PLOS Biology. Your manuscript has been evaluated by the PLOS Biology editors, by an Academic Editor with relevant expertise, and by four independent reviewers.

The reviews of your manuscript are appended below. You will see that the reviewers find the work potentially interesting. However, based on their specific comments and following discussion with the academic editor, I regret that we cannot accept the current version of the manuscript for publication. We remain interested in your study and we would be willing to consider resubmission of a comprehensively revised version that thoroughly addresses all the reviewers' comments. We cannot make any decision about publication until we have seen the revised manuscript and your response to the reviewers' comments. Your revised manuscript would be sent for further evaluation by the reviewers.

As you will see, the majority of reviewers found your work to be interesting and useful to the field. However, all of them raised several important concerns that we would need to see experimentally addressed before considering the manuscript further for publication. Among other issues, the reviewers ask that you confirm that the symbiosis is functional, provide evidence that your system works without false positives when other microbes are present and investigate whether there is a quantitative relationship between colonization and amount of pigment present. We would ask that you address reviewer #1’s concerns about the usefulness of the resource in the text, preferably describing how you see this being used as, indeed, it would seem to be a difficult tool to use in a wide variety of plant genetic backgrounds. The reviewers would also like the work to include information on how you plan to handle distribution of the resource you have created (seeds and DNA constructs). We suggest using a resource such as the Arabidopsis Biological Resource Center (https://abrc.osu.edu/researchers) or similar for distribution to improve the accessibility of your resource to the scientific community. 

We appreciate that these requests represent a great deal of extra work, and we are willing to relax our standard revision time to allow you six months to revise your manuscript. We expect to receive your revised manuscript within 6 months.

**IMPORTANT - SUBMITTING YOUR REVISION**

*Resubmission Checklist*

*Published Peer Review*

*PLOS Data Policy*

*Blot and Gel Data Policy*

Sincerely,

Aaron

Aaron Nicholas Bruns, Ph.D.,

Associate Editor,

abruns@plos.org,

PLOS Biology

REVIEWS:

Reviewer's Responses to Questions

Reviewer #1: This manuscript describes the development of betalains produced by transgenes under AM-specific promoters. The biggest constraint is that the construct will have to be introduced into every plant under examination. Whenever a different genetic background is needed, it must be introduced. In most species, such an introduction is a tedious process either by transformation or genetic crosses. So it doesn't easily lend itself to dissecting the genetic mechanism. It only allows for analysis in the same genetic background. In this reviewer's opinion, having to transform a large construct before the AM dynamics can be analyzed hardly counts as "significant outperformance" or offers the "potential to address, for the first time, a pressing biological question".

The authors are correct in stating that "The most powerful application of our betalain-based AM reporter lies in its ability to non-invasively document colonisation in fully developed root systems over time". They listed a number of possibilities. However, they failed to use their system to actually address any of these issues, and to demonstrate how they are pressing issues that cannot be addressed otherwise. 

Their claim that "A second important application is in the survey of induced or natural genetic variation that impacts on root system colonisation" is unconvincing. Not only one would have to introduce this reporter system into every line of a natural variation collection, transformation by itself will result in positional effect due to the randomness of transgene insertion site, or in their own words "varying degrees of transgene expression that could impact reporter intensity and functionality", which applies to variations between stably transformed lines as well as between hairy roots. 

My second concern is whether betalains are a robust reporter at all. Does it have enough resolution? In Medicago truncatula, the resolution seems rather low. The images are much more encouraging in N. benthamiana. It is hard to conclude which species is the outlier. Again, as the authors pointed out, the most promising application may be live imaging at the whole root system level. It is unclear why more conventional reporters, such as luciferase, would not suffice.

Reviewer #2: The manuscript by A. Timoneda et al. is a "method and resources" article that presents MycoRed, a method to visualize the occurrence of arbuscular mycorrhizal symbiosis in roots, in a non-destructive manner. The method is based on the expression of the betalain biosynthesis genes under control of AM symbiosis-specific promoters. 

The authors tested the systems in two dicots, the legume Medicago truncatula and Nicotiana benthamiana. They isolated AM symbiosis-specific promoters in N. benthamiana. They identified and tested constructs that are not expressed when the plants are not inoculated, which is essential for further use of the system (low false-positive rate). Lastly, they used a rhizotron system to show the potential for real-time monitoring of AM symbiosis. 

Symbiotic interactions are attracting more and more attention due to their potential as biofertilizer. This makes the development of such a tool particularly relevant and timely. As an expert in AM symbiosis research, I can foresee the potential for forward-genetic screens, which have been almost ignored due to technical difficulties, or to decipher cross interactions with biotic and abiotic factors. It is a much needed system and shows a great potential. 

I found the manuscript very clearly written and the reported experiments well conducted. 

As this article is submitted as a resource to be widely used by the community, I have, however, a few additional requests and suggestions.

Major requests:

- Confirm that the symbiosis is functional. Although the colonization of the transgenic lines, and the increased colonization rate observed over time, suggests that the association is "normal", it would be important to demonstrate it. As a proxy the authors should provide qRT-PCR data for the expression of marker genes such as RAM2 or STR/STR2 in the engineered lines following colonization by R. irregularis.

- Demonstrate the specificity. The authors tested the lines in artificial substrate and in vitro. The presented data are very convincing, however I am wondering whether the lines would behave similarly in more natural, microbe-rich, soils. The authors should test for the potential activation of betalain production in the "elite" line (NbPT5b-p3) in presence of microbes other than AM fungi. Including a pathogen that infect the roots would be particularly relevant. 

Suggestions:

i) The authors could provide a list (with reference numbers) for the L0 and L2 used for the study as well as a clear statement on the procedure to obtain these DNA modules, and .gb files of the vectors. 

ii) The authors could explain the procedure to obtain seeds of the stable N. benthamiana NbPT5b-p3, including the amount that can be possibly provided. Again, this is a resource and distribution is an important aspect.

iii) I was particularly impressed by the rhizotron experiment, because following AM symbiosis in real-time is a challenge that has never been solved. I was wondering whether the authors could use the images to quantify a proxy for the level of colonization, on the color-extracted files (red / white for instance?). This is obviously not a requirement, but having a way to quantify colonization with a simple scan, on living material, would be amazing. 

I am not an expert on the biosynthesis of Betalain and I did not evaluate that particular aspect of the manuscript. 

Reviewer #3: The manuscript by Timoneda and colleagues offers a nice technical contribution to the field of AM symbiosis. AM fungi develop inside the root tissues and there are only a few (and rare) morphological features that allow identifying the mycorrhizal root segments with the naked eye. For this reason, the development of transgenic plants which allow an easy detection of the arbuscules is surely a useful contribution.

The research has been carefully performed with clear experiments and correct controls. There are however some points which deserve attention, and some missing information should be added: all these issues have been listed following the text. In addition, the text sounds as a bit redundant since the same experiments are described for Medicago and then for Nicotiana, even if the results and the conclusions are very similar. I fully understand that the stable transformation of Nicotiana was done in a second moment, but probably a shortening of the text could lead to a more attractive reading. 

Title: is it clear that the visualisation of betalain pigments requires the introduction of reporter genes in model plants? In my opinion, the authors should state that a genetic transformation is at the basis of the experiment. 

Summary line 20: probably the word fungi after arbuscular mycorrhiza is missing. AM are not fungi!

Introduction

 Line 48. Glomeromycota. Please note that one of the most accepted taxonomy proposes AM fungi as Glomeromycotina, a subphylum (Spatafora et al 2016). Alternatively, see the taxonomic view as proposed by Leho Tedersoo 

Line 50 ".... can be formed by 70-90% of extant land plant species."

 Please note that the references are not the most updated. It is assumed that only 70-72% of land plants are colonized by AM fungi (see Brundett and Tedersoo, 2018; Genre et al 2020); 90% is the percentage of plants which are associated to mycorrhizal fungi. 

Line 57-60 Please note that also the other intracellular hyphae of AM fungi (those originated from the hyphopodia, the coils ..) are always surrounded by a host membrane. This new membrane is not exclusive of the arbuscules and is more correctly defined as the perifungal membrane. 

78: "complex microscopy " not clear

85-86 probably the yellow is not exclusive of cereals. Also Liliaceae have yellow mycorrhizal roots. 

96: Is there information on the natural expression of betalains in early diverging fungi?

143--144 ".... Heterologous expression of betalain biosynthesis genes specifically driven by AM-responsive promoters effectively tracked AM colonisation dynamics in both species.."

Perfect rationale, however, I would have appreciated to see the use of at least one gene also related to the first steps of colonization (i.e. signalling, transcription factors). These genes could give us more relevant information than the genes which are the markers of the established colonization 

Line 163: Can the authors provide some further information on the UBQ10 promoter? And provide some rational for the use of the constitutive promoters, which indeed have not been the better choice. 

Fig. 2 it seems that the betalain red is mostly associated to the inner root segments and not to the thinnest root branches (lateral roots). Have the authors done some statistical evaluation of the staining distribution looking at the morphology of the root system? 

Fig. 3 The red seems to be more abundant in the endodermis cells, where surely PT4 is not expressed. The authors write that red betacyanin distribution extends to cells beyond those with promoter activity ...Can betacyanin move across membranes, given that it is water-soluble? 

172: can the betalain expression change the transcriptomic profile of arbuscule containing cells?

204-206. Medicago is a model plant for many labs. It is not clear why the authors moved to Nicotiana instead of producing stable Medicago transformants. 

Fig. 4 Expression of NbPT5b and NbBCP1b increased after two weeks and showed significantly elevated transcript levels 3-4 weeks post inoculation (Fig 4a). In my opinion, this Fig 4 a does not show an original or unexpected result. I would suggest moving the Figure to the supplementary materials. Indeed the GUS constructs reveal a strong diffusion of the staining...

The two following paragraphs present many repetitions. Betalains can be used to visualise AM fungus colonisation in living Medicago truncatula roots vs Betalains visualise AM fungus colonisation in living Nicotiana benthamiana roots The authors should go in a more direct way to the solution, shortening the results obtained with the first constructs. It is clear that the constitutive promoters give problems... )

Fig. 5 a "Schematic of the multi-gene vectors constructed for inducible betalain expression in N. benthamiana roots where only the first gene of the betalain biosynthesis pathway is controlled by AM symbiosis specific promoters"

The legend is a bit confusing: the reader looks for differences between fig 5 vs. fig 2. But indeed the only difference is that promoters of CYP76AD1 are from Nicotiana and not from Medicago 

Fig 5 (e g) seems to be quite poor in quality. 

Table 1 the authors used ink staining to visualise the colonisation. But the method is not reported in the text (line 270) as well as the quantification methodology. Lastly, from which transformed plants were the data obtained?

Paragraph "Stable expression of NbPT5b-p1 and NbBCP1b-p1 can cause shoot developmental defects in N. benthamiana "

In my opinion this entire paragraph should be summarised, the details could be moved to the supplementary materials, since it is not so strictly relate to the main focus. On the other hand, it seems that the promoter PT5b is active in leaves. This could be an interesting result, even if not strictly related to this research. Is the transporter active in the leaves from Myc and not myc WT plants? It is known that PT is expressed also in root tips, irrespectively of the mycorrhizal colonization....

Fig. 6 b-e do not add any relevant information.... it could be fused with Fig. 7. Fig. 8 is much more interesting. In this context, it is not clear why the time course with the PT promoter is not shown, since it works better than the BCP one. 

Line 348-350 

Vegetative? Not clear...do they mean shoot and leaves? Also roots are vegetative organs. And at line 350 "… vegetative betalain production.."

364 what is the meaning? Why do they come back to medicago?

But.... does not make infection does not express PT4...since all the signalling is blocked and as a consequence the following processes...

The last part of the discussion (from 451 line) should be deleted since the point has already been largely introduced and discussed in the Results

Line 445: " A future solution to document total AM colonisation could involve the

Establishment of systems whereby the betalain biosynthesis genes are activated by transactivators, which could be then driven by promoters that are active at early, main and late

AM fungal colonisation stages (44-46). 

I apologize, but for me the sentence is not clear. Transactivators?

In conclusion, the authors should better show that the p3 constructs do not change plant and mycorrhizal phenotype, at least showing that the colonization percentage does not change between WT and transformants. 

Reviewer #4: The manuscript by Timoneda and coworkers presents the results of using Betalain biosynthetic genes to demonstrate effectiveness of the natural pigments as in vivo visual markers for arbuscular mycorrhizal colonization of both Medicago and tobacco root systems. They used AM-inducible promoters and assembled multi-gene reporter constructs and demonstrated that the innovative method MycoRed could allow for the non-invasive tracing of fungal colonization over time. The presented results fully support the conclusions and the study looks overall very convincing and solid. This research is of broad interest to the audience of plant biologists and timely places among an increasing interest and importance of investigating AM. 

The text reads fluently and the state of the art, results and conclusions are very clear. However, I have the following a few critics to raise.

Major points:

1. In this study, promoters of late AMS marker genes (MtPT4, MtBCP1, and tobacco homologs) were used to drive expression of betalain biosynthetic genes. The early infection and AM colonization is an essential part of AMS research, I would suggest testing the early AMS marker genes such as AM1 and AM3 and assay if the red pigmentation can be used to trace the infection at early stage of mycorrhizal infection, i.e., before 14 dpi. 

2. Stable transgenic tobacco plants were used to trace fungal colonization. A quantitative study could be performed to compare betalain pigmentation (by imaging or by HPLC) and fungal biomass accumulation (by qRT-PCR of R. irregularis housekeeping genes) and test if linear correlations exist. 

Minors:

Figure 3: Panels c and d, overlay images of pigments (red) and WGA-FITC staining (blue) 

should be provided.

Figure 5: Overlay images of panels d&e, f&g are recommended.

Figure 8: All dashed squares should be delimited in panel b.

Figure S9: Panel j, what are the two types of arrows?

Figure S11: Scale bars are missing.

Table 1: 4.8±13? 3.0±8? 1.1±3?

Table S1: 8.8±17? 2.4±7?

Line 235: Definition of NbEF is needed.

Line 263: d,f should be d,e

Line 266: d-d should be d-g.

Line 399: (a, b) should be (a, c)

Lines 571-574: space is need between digits and units, i.e., 40 mM instead of 40mM.

Line 591&592: NbEF and RiEF are inaccurate. Gene IDs or RefSeq should be provided.

---

## [Decision Letter · Decision Letter 2]

29 May 2021

Dear Dr Schornack,

Thank you for submitting your revised Methods and Resources manuscript entitled "MycoRed: Betalain pigments as in vivo real-time visual markers for arbuscular mycorrhizal colonisation of transgenic root systems" for publication in PLOS Biology. I have now obtained advice from three of the original reviewers and have discussed their comments with the Academic Editor, who has also checked your responses to the missing reviewer.

Based on the reviews, we will probably accept this manuscript for publication, provided you satisfactorily address the data and other policy-related requests included below. In addition, we would like you to consider a suggestion to improve the title, which is quite long:

"MycoRed: a versatile method for in vivo real-time visualization of arbuscular mycorrhizal colonisation of root systems"

We expect to receive your revised manuscript within two weeks. 

-  a cover letter that should detail your responses to any editorial requests.

*Published Peer Review History*

*Early Version*

Sincerely,

Ines

--

Ines Alvarez-Garcia, PhD,

Senior Editor,

ialvarez-garcia@plos.org,

PLOS Biology

Fig. 4A; Fig. S1; Fig. S8B and Fig. S10A, B

We require the original, uncropped and minimally adjusted images supporting all blot and gel results reported in an article's figures or Supporting Information files. We will require these files before a manuscript can be accepted so please prepare and upload them now. Please carefully read our guidelines for how to prepare and upload this data: https://journals.plos.org/plosbiology/s/figures#loc-blot-and-gel-reporting-requirements

Reviewers’ comments

Rev. 2: Pierre-Marc Delaux - note that this reviewer has signed his review.

In this revised version of their manuscript, the authors followed my three main suggestions:

1) They show convincing evidence that the expression of the betalain reporter does not impact symbiosis functionality. Although they only included data for Medicago it is reasonable to think that the same will be true for the Nb lines.

2) They provide additional evidence indicating the specificity of the reported (besides the fact that the substrate was not sterilized and thus contains other microorganisms).

3) They provide a detailled procedure to obtain the material, an essential aspect of this work submitted as a "Methods and Resources".

The reported conclusions are supported by the presented data and the work will be definitely an extremely useful resource for the plant symbiosis community. I do not have additional comments.

Rev. 3:

The Authors strongly improved their manuscript and solved some of the major criticisms. Even if not all the questions were faced, the rebuttal letter was fully convincing.

Rev. 4: Deqiang Duanmu - note that this reviewer has signed his review.

Accept.

---

## [Editor Report · Decision Letter 3]

16 Jun 2021

Dear Dr Schornack,

On behalf of my colleagues and the Academic Editor, Xinnian Dong, I am pleased to say that we can in principle offer to publish your Methods and Resources "MycoRed: Betalain pigments enable in vivo real-time visualization of arbuscular mycorrhizal colonization" in PLOS Biology, provided you address any remaining formatting and reporting issues. These will be detailed in an email that will follow this letter and that you will usually receive within 2-3 business days, during which time no action is required from you. Please note that we will not be able to formally accept your manuscript and schedule it for publication until you have made the required changes.

PRESS

Sincerely, 

Ines

--

Ines Alvarez-Garcia, PhD 

Senior Editor 

PLOS Biology
